# GRAPH DOMAIN ADAPTATION VIA THEORY-GROUNDED SPECTRAL REGULARIZATION

**Yuning You**[1]**, Tianlong Chen**[2]**, Zhangyang Wang**[2]**, Yang Shen**[1]

[1]Department of Electrical and Computer Engineering, Texas A&M University
[2]Department of Electrical and Computer Engineering, University of Texas at Austin
{yuning.you,yshen}@tamu.edu, {tianlong.chen,atlaswang}@utexas.edu

## ABSTRACT

Transfer learning on graphs drawn from varied distributions (domains) is in great demand across many applications. Emerging methods attempt to learn domain-invariant representations using graph neural networks (GNNs), yet the empirical performances vary and the theoretical foundation is limited. This paper aims at designing theory-grounded algorithms for graph domain adaptation (GDA). (**i**) As the first attempt, we derive a model-based GDA bound closely related to two GNN spectral properties: *spectral smoothness (SS)* and *maximum frequency response (MFR)*. This is achieved by cross-pollinating between the OT-based (optimal transport) DA and graph filter theories. (**ii**) Inspired by the theoretical results, we propose algorithms regularizing spectral properties of SS and MFR to improve GNN transferability. We further extend the GDA theory into the more challenging scenario of conditional shift, where spectral regularization still applies. (**iii**) More importantly, our analyses of the theory reveal which regularization would improve performance of what transfer learning scenario, (**iv**) with numerical agreement with extensive real-world experiments: SS and MFR regularizations bring more benefits to the scenarios of *node transfer* and *link transfer*, respectively. In a nutshell, our study paves the way toward explicitly constructing and training GNNs that can capture more transferable representations across graph domains. Codes are released at https://github.com/Shen-Lab/GDA-SpecReg.

## 1 INTRODUCTION

Many applications call for "transferring" graph representations learned from one distribution (domain) to another, which we refer to as graph domain adaptation (GDA). Examples include temporally-evolved social networks (Wang et al., 2021), molecules of different scaffolds (Hu et al., 2019), and protein-protein interaction networks in various species (Cho et al., 2016). In general, this setting of transfer learning is challenging due to the data-distribution shift between the training (source) and test (target) domains (i.e. $\mathbb{P}_S(G, Y) \neq \mathbb{P}_T(G, Y)$). In particular, such a challenge escalates for graph-structured data that are abstractions of diverse nature (You et al., 2021; 2022).

Despite the tremendous needs arising from real-world applications, current methods for GDA (as reviewed in Section 2) mostly fall short in delivering competitive target performance *with theoretical guarantee*. Inevitably those approaches assuming distribution invariance (or adopting heuristic principles) are restricted in theory (Garg et al., 2020; Verma & Zhang, 2019). The emerging approaches (Zhang et al., 2019; Wu et al., 2020) straightforwardly apply adversarial training between source and target representations, intentionally founded on the DA theory to bound the target risk (Redko et al., 2020). However, the generic DA bound in theory is agnostic to graph data and models, which could be more precisely tailored for graphs. We therefore set out to explore the following question: *How to design algorithms to boost transfer performance across different graph domains, with the grounded theoretical foundation?* Our step-by-step answers are as follows.

(**i**) **Derivation of model-based GDA bound.** Building upon the rigorous assurance established in the DA theory (Section 3), we start by directly rewriting the OT-based (optimal transport) DA bound (Redko et al., 2017; Shen et al., 2018) for graphs (Corollary 1), which is closely coupled with the *Lipschitz* constant of graph encoders. The nontrivial challenge here is how to formulate GNN Lipschitz w.r.t the distance metric of non-Euclidean data. Leveraging the graph filter theory (Gama et al., 2020; Arghal et al., 2021), we first state that GNNs can be constructed stably w.r.t. the

misalignment of edges and that of node features, multiplied by two spectral properties respectively: *spectral smoothness (SS)* and *maximum frequency response (MFR)* (Lemma 1). Subsequently, we utilize SS and MFR to formulate GNN Lipschitz w.r.t graph distances into a general form, and instantiate it as (informally) $\max\{\mathcal{O}(\mathrm{SS}), \mathcal{O}(\mathrm{MFR})\}$ w.r.t. the commonly-used matching distance (Gama et al., 2020; Arghal et al., 2021) (Lemma 2). This leads to the first model-based GDA bound.

(ii) **Theory-grounded spectral regularization.** One potential way to tighten the DA bound is to modulate the Lipschitz constant (Section 3). Guided by the theoretical results above, we are well-motivated to propose spectral regularization (i.e. SSReg and MFRReg) to restrict the target risk bound (Section 4.2). We also extend the GDA theory into the more challenging conditional-shift scenario (Li et al., 2021a; Zhao et al., 2019) (Lemma 3), where spectral regularization still applies.

(iii) **Interpretation on how theory drives practice.** Our further analyses on theory reveal *which regularization would improve performance of what graph transfer scenario*: specifically, SSReg and MFRReg are respectively beneficial to the scenarios of *node transfer* and *link transfer* (Section 4.2), (iv) **with extensive numerical evidences** from un/semi-supervised (cross-species protein-protein interaction) link prediction and (temporally-shifted paper topic) node classification (Section 5).

## 2 RELATED WORKS

**Self-supervision on graphs.** Graph self-supervised learning, surging recently, learns empirically more generalizable representations through exploiting vast unlabelled graph data (please refer to (Xie et al., 2021) for a comprehensive review). The success of self-supervision largely hinges on big data and, more importantly, heuristically-designed pretext tasks. The tasks can be predictive (Velickovic et al., 2019; Hu et al., 2019; Jin et al., 2020; You & Shen, 2022; You et al., 2020b; Chien et al., 2021; Talukder et al., 2022) or contrastive (You et al., 2020a; Zhu et al., 2020b; Qiu et al., 2020; Wei et al., 2022), which does not provide theoretical guarantee of the target performance and, as a result, occasionally leads to "negative transfer" in practice (Hu et al., 2019; You et al., 2020a).

**Transferring GNNs with explicit covariate shifts.** To promote target performance, one line of work is to utilize more data and make specific assumptions. One such example is to assume the access to source labels and the explicit covariate shift that $\mathbb{P}_S(Y|G) = \mathbb{P}_T(Y|G)$ and $\mathbb{P}_S(G) \neq \mathbb{P}_T(G)$ in a specific way, which enables theoretical tools for certain guarantees. (Ruiz et al., 2020; Yehudai et al., 2021) study the specific setting of size generalization and use the graphon theory (Lovász, 2012) to develop size-invariant representations. (Bevilacqua et al., 2021) works on transfer learning in shifting $d$-patterns of subgraphs and adopts the theory of GNN expressiveness (Xu et al., 2018; Morris et al., 2019) to demonstrate the existence of negative-transferring GNNs despite their universal approximation capability. Accordingly, the study proposes $d$-pattern classification pre-training to help escape from negative-transferring GNNs. These methods are restricted to the designated transfer learning scenarios. Besides, some other works (Fan et al., 2021; Sui et al., 2021; Li et al., 2021b; Kenlay et al., 2021; Chen et al., 2022; Li et al., 2022; Zhang et al., 2022; Jin et al., 2022) adopt the implicit covariate shift assumption with source labels while lacking assurance in theory, e.g. (Wu et al., 2022a;b) assumes that the shift could be implicitly modeled with an environment learner (please refer to (Gui et al., 2022) for a comprehensive review).

**Graph domain adaptation.** To deliver a generally applicable guarantee, several methods (Dai et al., 2019; Cai et al., 2021; Zhang et al., 2019; Wu et al., 2020; Xu et al., 2022) additionally utilize target graphs to learn domain-invariant representations. According to the DA theory (Ben-David et al., 2007; 2010; Redko et al., 2020; Zhang et al., 2020; Yan et al., 2017), the target risk is guaranteed to be bounded (please refer to (Redko et al., 2020) for a comprehensive review). The generic DA bound is not designated for graph data or encoders where further improvement could be achieved.

## 3 PRELIMINARIES

**Problem setup.** We are given i.i.d. samples (Verma & Zhang, 2019; Zhu et al., 2021; Cong et al., 2021) and their labels $\{(G_n, Y_n)\}_{n=1}^{N_S}$ from the source distribution $\mathbb{P}_S(G, Y)$ of graphs $G \in \mathcal{G}$ and labels $Y \in \mathcal{Y}$, where $G = \{V, E\}$ is associated with the set of nodes $V$ and edges $E$, together with the node feature $X \in \mathcal{R}^{|V| \times D}$ and adjacency matrices $A \in \mathcal{R}^{|V| \times |V|}$. We also have access to unlabeled samples $\{G_n\}_{n=1}^{N_T}$ from the marginalized target distribution $\int \mathbb{P}_T(G, Y) dY$. With the covariate shift assumption that $\mathbb{P}_S(G) \neq \mathbb{P}_T(G), \mathbb{P}_S(Y|G) = \mathbb{P}_T(Y|G)$ (Ben-David et al., 2007; 2010), we are expected to train a graph neural network (GNN) $h : \mathcal{G} \to \mathcal{Y}$ with the accessible data and then evaluate on target samples from $\mathbb{P}_T(G, Y)$.

**Domain adaptation with optimal transport.** Studies on transfer learning across distinctly distributed data have proliferated in the past few years, known as domain adaptation (DA) (Redko et al., 2020). Based on the aforementioned problem setup, we decompose the trained GNN $h = g \circ f$ into the feature extractor $f : \mathcal{G} \to \mathcal{R}^{D'}$ ($Z = f(G)$) and discriminator $g : \mathcal{R}^{D'} \to \mathcal{Y}$ ($Y = g(Z)$). Without the loss of generalizability, we consider a binary classification task where $\mathcal{Y} = [0, 1]$ and $Y \in \mathcal{Y}$ is the probability to belong to class 1. We denote the labeling function given representations as $\hat{g} : \mathcal{R}^{D'} \to \mathcal{Y}$, and the (empirical) source and target risks as $\hat{\epsilon}_S(g, \hat{g}) = \frac{1}{N_S} \sum_{n=1}^{N_S} |g(Z_n) - \hat{g}(Z_n)|$ and $\epsilon_T(g, \hat{g}) = \mathbb{E}_{\mathbb{P}_T(Z)}\{|g(Z) - \hat{g}(Z)|\}$, respectively. Applying DA with optimal transport (OT), if the covariate shift holds on representations that $\mathbb{P}_S(Y|Z) = \mathbb{P}_T(Y|Z)$, the target risk $\epsilon_T(g, \hat{g})$ is bounded as in the following theorem.

**Theorem 1 (Redko et al., 2017; Shen et al., 2018; Li et al., 2021a)** Suppose that the learned discriminator $g$ is $C_g$-Lipschitz where the Lipschitz norm $\|g\|_{\text{Lip}} = \max_{Z_1, Z_2} \frac{|g(Z_1) - g(Z_2)|}{\rho(Z_1, Z_2)} = C_g$ holds for some distance function $\rho$ (Euclidean distance here). Let $\mathcal{H} := \{g : \mathcal{Z} \to \mathcal{Y}\}$ be the set of bounded real-valued functions with the pseudo-dimension $Pdim(\mathcal{H}) = d$ that $g \in \mathcal{H}$, with probability at least $1 - \delta$ the following inequality holds:

$$\epsilon_T(g, \hat{g}) \leq \hat{\epsilon}_S(g, \hat{g}) + \sqrt{\frac{4d}{N_S} \log(\frac{eN_S}{d}) + \frac{1}{N_S} \log(\frac{1}{\delta})} + 2C_g W_1\Big(\mathbb{P}_S(Z), \mathbb{P}_T(Z)\Big) + \omega, \quad (1)$$

where $\omega = \min_{\|g\|_{\text{Lip}} \leq C_g} \{\epsilon_S(g, \hat{g}) + \epsilon_T(g, \hat{g})\}$ denotes the model discriminative ability (to capture source and target data), and the first Wasserstein distance is defined as (Villani, 2009):

$$W_1(\mathbb{P}, \mathbb{Q}) = \sup_{\|g\|_{\text{Lip}} \leq 1} \Big\{\mathbb{E}_{\mathbb{P}_S(Z)} g(Z) - \mathbb{E}_{\mathbb{P}_T(Z)} g(Z)\Big\}. \quad (2)$$

The thorough *tightness* justification of the OT-based DA bound can be found in ((Redko et al., 2020), Section 5.3-5.5). The theorem indicates that the generalization gap depends on both the domain-divergence $2C_g W_1(\mathbb{P}_S(Z), \mathbb{P}_T(Z))$ and the model discriminability $\omega$.

**Adversarial training.** Motivated from Theorem 1 (or its variants differing mainly in distribution divergence metrics (Ben-David et al., 2007; 2010; Mansour et al., 2009)), a well-developed practice is to learn domain-invariant representations via jointly optimizing the source risk and the distribution divergence term (conceptually, $W_1(\mathbb{P}_S(Z), \mathbb{P}_T(Z))$) as in (Redko et al., 2017; Shen et al., 2018):

$$\min_{f, \|g\|_{\text{Lip}} \leq C_g} \frac{1}{N_S} \sum_{n=1}^{N_S} \ell(g \circ f(G_n), Y_n) + \gamma \hat{W}_1\Big(\mathbb{P}_S(f(G)), \mathbb{P}_T(f(G))\Big), \quad (3)$$

where $\ell$ is the loss function used for training, $\hat{W}_1$ is the empirically calculated first Wasserstein distance with implementation details following (Shen et al., 2018) presented in Appendix E, and $\gamma$ is the trade-off factor. Besides co-optimizing $\hat{W}_1$, implementing domain classifiers is another effective way to alleviate the domain discrepancy (Zhang et al., 2019; Wu et al., 2020).

## 4 METHODS

### 4.1 MODEL-BASED DA BOUND FOR GRAPH-STRUCTURED DATA

Noticing that the generic DA theory (Theorem 1) is agnostic to data structures and encoders, our first step is to directly *rewrite* it for graph-structured data ($G$) accompanied with graph feature extractors ($f$) as follows. The *covariate shift assumption* is now reframed as $\mathbb{P}_S(Y|G) = \mathbb{P}_T(Y|G)$.

**Corollary 1.** Let's assume that the learned discriminator is $C_g$-Lipschitz continuous as described in Theorem 1, and the graph feature extractor $f$ (also referred to as GNN) is $C_f$-Lipschitz that $\|f\|_{\text{Lip}} = \max_{G_1, G_2} \frac{\|f(G_1) - f(G_2)\|_2}{\eta(G_1, G_2)} = C_f$ for some graph distance measure $\eta$. Let $\mathcal{H} := \{h : \mathcal{G} \to \mathcal{Y}\}$ be the set of bounded real-valued functions with the pseudo-dimension $Pdim(\mathcal{H}) = d$ that $h = g \circ f \in \mathcal{H}$, with probability at least $1 - \delta$ the following inequality holds:

$$\epsilon_T(h, \hat{h}) \leq \hat{\epsilon}_S(h, \hat{h}) + \sqrt{\frac{4d}{N_S} \log(\frac{eN_S}{d}) + \frac{1}{N_S} \log(\frac{1}{\delta})} + 2C_f C_g W_1\Big(\mathbb{P}_S(G), \mathbb{P}_T(G)\Big) + \omega, \quad (4)$$

where the (empirical) source and target risks are $\hat{\epsilon}_S(h, \hat{h}) = \frac{1}{N_S} \sum_{n=1}^{N_S} |h(G_n) - \hat{h}(G_n)|$ and $\epsilon_T(h, \hat{h}) = \mathbb{E}_{\mathbb{P}_T(G)}\{|h(G) - \hat{h}(G)|\}$, respectively, where $\hat{h} : \mathcal{G} \to \mathcal{Y}$ is the labeling function for graphs and $\omega = \min_{\|g\|_{\text{Lip}} \leq C_g, \|f\|_{\text{Lip}} \leq C_f} \{\epsilon_S(h, \hat{h}) + \epsilon_T(h, \hat{h})\}$.

The property in Corollary 1 designated for GNNs, as we focus on, is the Lipschitz constant $C_f = \max_{G_1, G_2} \frac{\|f(G_1) - f(G_2)\|_2}{\eta(G_1, G_2)}$. Success in instantiating this conceptual *model-related* property for graphs provides important insights into how to rationally construct or train GNNs for better transferability. Other data-relevant properties (e.g. $W_1(\mathbb{P}_S(G), \mathbb{P}_T(G))$) are left to future works.

Instantiating the GNN Lipschitz constant is however nontrivial. The distance metric $\eta(G_1, G_2)$ is formulated w.r.t non-Euclidean structures and its computation hinges on solving the graph matching problem whose time complexity is exponential in the number of nodes (Riesen & Bunke, 2009; Riesen et al., 2010).

Rather than working on the denominator of the Lipschitz norm, we take an alternative perspective of the numerator, i.e. $\|f(G_1) - f(G_2)\|_2$, which is related to the GNN stability (potentially multiplied with the distance term to eliminate the denominator). This essentially motivates us to draw a connection between transferability and stability. Thanks to the permutation invariance property of GNNs, graph matching does not need to be solved here. Building upon the graph filter theory (Gama et al., 2020; Arghal et al., 2021), we first state that GNNs can be constructed stably:

☐ **Lemma 1.** Suppose that $\mathcal{G}$ is the set for graphs of the size the size $N_G$ after padding with isolated nodes, similar to (Zhu et al., 2021). Given $\forall G_1, G_2 \in \mathcal{G}$ and $A_1 = U_1 \Lambda_1 U_1^\mathsf{T}, A_2 = U_2 \Lambda_2 U_2^\mathsf{T}$, the eigen decomposition for adjacency matrices $A_1$ and $A_2$ that $\Lambda_1 = \text{diag}([\lambda_{1,1}, ..., \lambda_{1,N_G}]), \Lambda_2 = \text{diag}([\lambda_{2,1}, ..., \lambda_{2,N_G}])$ (eigen values are sorted in the descending order). A GNN is constructed by composing a graph filter and nonlinear mapping that $f(G_1) = r(\sigma(\mathcal{S}(A_1) X_1 W)) = r(\sigma(U_1 \mathcal{S}(\Lambda_1) U_1^\mathsf{T} X_1 W))$ where $\mathcal{S}$ is the polynomial function that $\mathcal{S}(A_1) = \sum_{k=0}^{\infty} s_k A_1^k$, $W \in \mathcal{R}^{D \times D'}$ is the learnable weight matrix, $r$ is the mean/sum/max readout function to pool node representations, and the pointwise nonlinearity holds as $|\sigma(b) - \sigma(a)| \le |b - a|, \forall a, b \in \mathcal{R}$. Assuming $\|X\|_{\text{op}} \le 1$ and $\|W\|_{\text{op}} \le 1$ ($\|\cdot\|_{\text{op}}$ stands for operator norm), the following inequality holds:

$$\|f(G_1) - f(G_2)\|_2 \le C_\lambda (1 + \tau \sqrt{N_G}) \|A_1 - P^* A_2 P^{*\mathsf{T}}\|_\mathsf{F}$$
$$+ \mathcal{O}(\|A_1 - P^* A_2 P^{*\mathsf{T}}\|_\mathsf{F}^2) + \max\left\{|\mathcal{S}(\Lambda_2)|\right\} \|X_1 - P^* X_2\|_\mathsf{F}, \tag{5}$$

where $\tau = (\|U_1 - U_2\|_\mathsf{F} + 1)^2 - 1$ stands for the eigenvector misalignment which can be bounded, $P^* = \text{argmin}_{P \in \Pi} \{\|X_1 - P X_2\|_\mathsf{F} + \|A_1 - P A_2 P^\mathsf{T}\|_\mathsf{F}\}$, $\Pi$ is the set of permutation matrices, $\mathcal{O}(\|A_1 - P^* A_2 P^{*\mathsf{T}}\|_\mathsf{F}^2)$ is the remainder term with bounded multipliers defined in (Gama et al., 2020), and $C_\lambda$ is the spectral Lipschitz constant that $\forall \lambda_i, \lambda_j, |\mathcal{S}(\lambda_i) - \mathcal{S}(\lambda_j)| \le C_\lambda |\lambda_i - \lambda_j|$.

*Proof.* See Appendix A.

The lemma integrates the stability properties w.r.t. node feature and edge perturbations in (Gama et al., 2020) and (Arghal et al., 2021), respectively, where $C_\lambda$ tied to edge perturbations characterizes *spectral smoothness (SS)* of the underlining graph filter, and $\max(|\mathcal{S}(\Lambda_2)|)$ tangled with node feature perturbations defines *maximum frequency response (MFR)*. We note that in practice for Lemma 1, no padding is actually needed if the summation pooling is adopted as in our experiments (Xu et al., 2018), since the padded nodes do not contribute to the graph embedding (Zhu et al., 2021). Analysis on more sophisticated architectures is left to future works.

We next instantiate the GNN Lipschitz constant for the commonly-used matching distance (Gama et al., 2020; Arghal et al., 2021) in the following lemma, achieved by eliminating the distance term in the numerator and denominator of the GNN Lipschitz constant.

☐ **Lemma 2.** Suppose that $\mathcal{G}$ is the set for graphs of the size $N_G$ after padding with isolated nodes, similar to (Zhu et al., 2021). We define the matching distance between $G_1, G_2 \in \mathcal{G}$ as $\eta(G_1, G_2) = \min_{P \in \Pi} \{\|X_1 - P X_2\|_\mathsf{F} + \|A_1 - P A_2 P^\mathsf{T}\|_\mathsf{F}\}$. Suppose that the edge perturbation is bounded that $\forall G_1, G_2 \in \mathcal{G}, \|A_1 - P^* A_2 P^{*\mathsf{T}}\|_\mathsf{F} \le \varepsilon$ with the optimal permutation $P^*$, and there exists an eigenvalue $\lambda^* \in \mathcal{R}$ to achieve the maximum $|\mathcal{S}(\lambda^*)| < \infty$. We can then calculate the Lipschitz constant of GNN as:

$$C_f = \max\left\{C_\lambda K_1 + \varepsilon K_2, |\mathcal{S}(\lambda^*)|\right\}, \tag{6}$$

where $K_1, K_2$ is the supremes of $(1 + \tau \sqrt{N_G})$ and the remainder multiplier in Lemma 1, respectively, following similar philosophies in ((Gama et al., 2020), Theorem 1).

*Proof.* See Appendix B.

To recap, $C_\lambda$ and $|\mathcal{S}(\lambda^*)|$ characterize SS and MFR, respectively. Thereafter, *the direct incorporation of Corollary 1 and Lemma 2 results in the first model-based DA bound for graph data.* It builds the foundation to understand and further develop GDA algorithms. Although in our theory GNN is assumed to be composed of a graph filter and a nonlinear activation layer, in practice such an architecture is ubiquitous, well-known for its simplicity and effectiveness (Wu et al., 2019; Li et al., 2019). We also provide theory for two-layer GNNs in Appendix I which is naturally extendable to multi-layer architectures via induction.

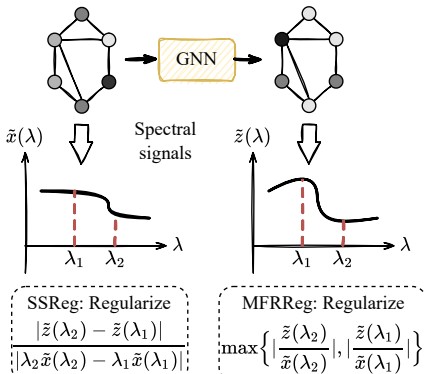

**Figure 1:** An overview of spectral regularization. In implementation we regularize on the subtraction between numerators and denominators multiplied by certain threshold, considering the numerical instability issue in division.

## 4.2 THEORY-GROUNDED SPECTRAL REGULARIZATION

Recall that, in the DA inequality (4) (see also Section 3), the gap between the target and source risks is bounded with (**i**) the distribution divergence between source and target ($W_1$ term) *multiplied by* Lipschitz constants, and (**ii**) the discriminative capability of NNs to capture invariant knowledge (the $\omega$ term) *restricted by* Lipschitz constants. This indicates that varying the Lipschitz constant results in the intrinsic *trade-off* between domain-divergence and discriminability to vary the DA bound.

Therefore, in search of a sweet spot in the trade-off, one way to tighten the bound is to regularize the Lipschitz constant of NNs. Equipped with the derived graph model-based DA bound in Section 4.1, we are now ready to propose regularizing the GNN spectral properties of SS or MFR for balancing between domain-divergence and discriminability. We implement regularization as follows.

**Implementations.** Driven by the analysis, we incorporate spectral regularization (SpecReg $\in$ {SSReg, MFRReg}, SSReg for $C_\lambda$ and MFRReg for $|\mathcal{S}(\lambda^*)|$ in equation (6)) into the traditional domain-invariant representation learning framework (3) as follows:

$$\min_{f, \|g\|_{\mathrm{Lip}} \leq C_g} \frac{1}{N_\mathrm{S}} \sum_{n=1}^{N_\mathrm{S}} \ell(g \circ f(G_n), Y_n) + \gamma \hat{W}_1 \Big( \mathbb{P}_\mathrm{S}(f(G)), \mathbb{P}_\mathrm{T}(f(G)) \Big)$$
$$+ \gamma' \mathrm{SpecReg}\Big( f, \{G_n\}_{n=1}^{N_\mathrm{S}}, \{G_n\}_{n=1}^{N_\mathrm{T}} \Big), \qquad (7)$$

where $\gamma'$ is the trade-off factor, tuned over {1, 1e-1, 1e-2, 1e-3} through validation. Denote $Z = f(G)$ and the spectral signals $\tilde{Z} = U^\mathsf{T} Z$, $\tilde{X} = U^\mathsf{T} X$, SS and MFR regularization (SSReg, MFRReg) are specifically implemented by regularizing on the spectral outputs w.r.t. spectral inputs as:

$$\mathrm{SSReg}\Big( f, \{G_n\}_{n=1}^{N_\mathrm{S}}, \{G_n\}_{n=1}^{N_\mathrm{T}} \Big) = \sum_{\mathrm{D} \in \{\mathrm{S,T}\}} \frac{1}{N_\mathrm{D}} \sum_{n=1}^{N_\mathrm{D}} \mathrm{sum}\Big( \mathrm{ReLU}(\mathrm{abs}(\tilde{Z}_n[2:N_G,:]$$
$$- \tilde{Z}_n[1:N_G-1,:]) - \upsilon \, \mathrm{abs}(\Lambda_n \tilde{X}_n[2:N_G,:] - \Lambda_n \tilde{X}_n[1:N_G-1,:]))\Big), \quad (8)$$

$$\mathrm{MFRReg}\Big( f, \{G_n\}_{n=1}^{N_\mathrm{S}}, \{G_n\}_{n=1}^{N_\mathrm{T}} \Big) = \sum_{\mathrm{D} \in \{\mathrm{S,T}\}} \frac{1}{N_\mathrm{D}} \sum_{n=1}^{N_\mathrm{D}} \mathrm{sum}\Big( \mathrm{ReLU}(\mathrm{abs}(\tilde{Z}_n) - \upsilon \, \mathrm{abs}(\tilde{X}_n))\Big),$$
$$(9)$$

where $X[m,n]$ denotes matrix indexing, sum is the matrix summation, ReLU and abs are pointwise rectifier and absolute functions, respectively, and $\upsilon$ is the threshold value tuned over {0.1, 1, 10, 100, 1000} through validation. An overview of the implementation is depicted in Figure 1. Analysis on complexity of spectral regularization (with eigendecomposition) is presented in Appendix H.

**How could theory further drive practice?** We also notice that in special scenarios, either of the two regularizations (SS and MFR) can be favored than the other. (**i**) *Node transfer*: When the node features preserve (mostly) the invariant label-related information, while edges are less label-relevant (noisy) and distribute highly distinctly across source and target domains, we would have

large $K_1, \varepsilon$ in Lemma 2 that $C_\lambda K_1 + \varepsilon K_2 > |\mathcal{S}(\lambda^*)|$. Thus regularizing SS leads to the decreased domain-divergence term ($C_f$ decreases) and the less increased discriminability term w.r.t. node features (GNN is sensitive enough to label-preserving node features as shown in Lemma 1). **(ii)** *Link transfer*: Oppositely, when edges are invariantly label-preserving across source and target domains whereas node features are relatively noisy, regularizing MFR is more desired. This provides us with the guidance to select the regularization in practical applications with domain knowledge.

### 4.3 EXTENSION TO THE SEMI-SUPERVISED SETTING

Results so far do not assume the access to target labels, which is in the unsupervised setting. In this section, we show that our analysis also applies in the semi-supervised setting, where the more challenging conditional shift is considered—$\mathbb{P}_S(Y|G) \neq \mathbb{P}_T(Y|G)$—in addition to the access to a small amount of target labelled data (Li et al., 2021a; Zhao et al., 2019). We provide a finite-sample semi-supervised OT-based GDA bound in the following lemma.

☐ **Lemma 3.** Under the assumption of Corollary 1, we further assume that there exists a small amount of i.i.d. samples with labels $\{(G_n, Y_n)\}_{n=1}^{N_T'}$ from the target distribution $\mathbb{P}_T(G, Y)$ ($N_T' \ll N_S$) and bring in the conditional shift assumption that domains have different labeling function $\hat{h}_S \neq \hat{h}_T$ and $\max_{G_1, G_2} \frac{|\hat{h}_D(G_1) - \hat{h}_D(G_2)|}{\eta(G_1, G_2)} = C_h \leq C_f C_g$ (D $\in \{S, T\}$) for some constant $C_h$ and distance measure $\eta$. Let $\mathcal{H} := \{h : \mathcal{G} \to \mathcal{Y}\}$ be the set of bounded real-valued functions with the pseudo-dimension $Pdim(\mathcal{H}) = d$, with probability at least $1 - \delta$ the following inequality holds:

$$\epsilon_T(h, \hat{h}_T) \leq \frac{N_T'}{N_S + N_T'} \hat{\epsilon}_T(h, \hat{h}_T) + \frac{N_S}{N_S + N_T'} \hat{\epsilon}_S(h, \hat{h}_S) + \frac{N_S}{N_S + N_T'} \left( \left( \frac{8d}{N_T'} \log(\frac{eN_T'}{d}) + \frac{2}{N_T'} \log(\frac{1}{\delta}) \right. \right.$$

$$\left. \left. + \frac{8d}{N_S} \log(\frac{eN_S}{d}) + \frac{2}{N_S} \log(\frac{1}{\delta}) \right)^{\frac{1}{2}} + 2C_f C_g W_1(\mathbb{P}_S(G), \mathbb{P}_T(G)) + \omega \right), \quad (10)$$

where $\omega = \min\left( |\epsilon_S(h, \hat{h}_S) - \epsilon_S(h, \hat{h}_T)|, |\epsilon_T(h, \hat{h}_S) - \epsilon_T(h, \hat{h}_T)| \right)$.

*Proof.* See Appendix C.

Since the main form of inequality (10) is consistent with inequality (4), similar discussion can be made following the thoughts in Section 4.2, to demonstrate that we can properly regularize the spectral properties of GNNs (SS and MFR) to tighten the target risk and seek a sweet spot between the domain-divergence and discriminability. Following optimization (7), the semi-supervised training procedure is implemented as:

$$\min_{f, \|g\|_{\text{Lip}} \leq C_g} \frac{1}{N_S} \sum_{n=1}^{N_S} \ell(g \circ f(G_n), Y_n) + \frac{1}{N_T'} \sum_{n=1}^{N_T'} \ell(g \circ f(G_n), Y_n) + \gamma \hat{W}_1\left( \mathbb{P}_S(f(G)), \mathbb{P}_T(f(G)) \right)$$

$$+ \gamma' \text{SpecReg}\left( f, \{G_n\}_{n=1}^{N_S}, \{G_n\}_{n=1}^{N_T} \right). \quad (11)$$

## 5 EXPERIMENTS

We evaluate our proposed algorithms, SSReg and MFRReg, in two real-world applications of graph transfer learning : **(i)** link prediction of protein-protein interactions (PPIs) across different species (Szklarczyk et al., 2021), and **(ii)** node classification of paper topics across different time periods (Wu et al., 2020; Hu et al., 2020).

### 5.1 PREDICTING PROTEIN-PROTEIN INTERACTIONS IN VARIOUS SPECIES

**Datasets.** PPI networks have proven important to understand functional genomics and analyze biological pathways (Sharan et al., 2007; Navlakha & Kingsford, 2010). But in most species the coverage of experimental PPI data remains low (Sledzieski et al., 2021). We utilize protein sequences together with freely accessible computational PPIs via whole-genome comparisons (Szklarczyk et al., 2021) to predict experimental PPIs, i.e. the graph is built with nodes represented as protein sequences and edges as computational PPIs. We collect PPIs of species from the STRING database (Szklarczyk et al., 2021) where PPIs in the *neighborhood*, *fusion* and *co-occurrence* channels are defined computational, and in the

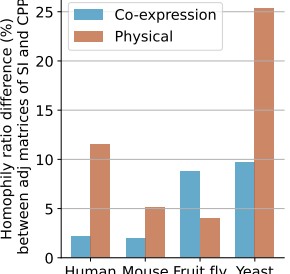

**Figure 2:** Homophily ratio difference between adjacency matrices built with sequence identity (SI) and computational PPI (CPPI). Higher values indicate more label-preserving of protein sequences.

**Table 1:** Unsupervised transfer of cross-species protein-protein co-expression interaction prediction. Numbers in red are the top-2 AUROC (AUPRC) (mean±std%). DA-C, DA-W denotes domain-invariant representation learning with domain classifiers (Ganin et al., 2016) and guided by Wasserstein distance as in optimization (3) (Redko et al., 2017; Shen et al., 2018), respectively.

| Methods | Co-expression: Link transfer ⇒ MFRReg | | | | | |
| | Mouse | Zebrafish | Fruit fly | Yeast | Mean↑ | Rank↓ |
| --- | --- | --- | --- | --- | --- | --- |
| Mashup | 48.98±2.34 | 51.63±1.81 | 50.28±2.20 | 46.31±0.63 | 43.90 | 9.0 |
| | (5.49±0.32) | (5.37±0.33) | (5.51±0.17) | (4.96±0.05) | (5.33) | |
| D-SCRIPT | 54.48±3.27 | 61.18±1.05 | 66.63±1.41 | 58.88±0.80 | 60.29 | 7.5 |
| | (6.01±0.63) | (8.12±1.77) | (9.78±0.25) | (7.53±0.11) | (7.86) | |
| GraphCL | 73.09±1.56 | 74.19±0.50 | 66.80±2.55 | 62.41±1.12 | 69.12 | 5.2 |
| | (14.98±2.18) | (18.76±2.11) | (12.12±2.00) | (11.32±2.86) | (14.29) | |
| Transformer | 69.55±0.41 | 69.63±0.84 | 57.38±1.77 | 63.01±1.45 | 64.89 | 5.8 |
| | (18.06±0.13) | (27.44±1.21) | (10.13±1.02) | (11.25±1.58) | (16.72) | |
| Transformer +GIN | 76.35±0.38 | 79.29±2.78 | 66.54±1.11 | **63.91**±1.55 | 71.52 | 4.0 |
| | (21.91±1.60) | (28.07±4.71) | (13.48±0.71) | (11.15±1.11) | (18.65) | |
| Transformer +GIN+DA-C | **78.56**±1.55 | **79.46**±2.97 | 64.78±1.23 | 60.65±3.85 | 70.86 | 4.8 |
| | (22.76±4.42) | (27.10±3.10) | (11.61±2.08) | (10.72±2.44) | (18.04) | |
| Transformer +GIN+DA-W | 77.38±2.54 | 79.22±0.89 | **67.78**±0.40 | 62.43±2.62 | **71.70** | 3.6 |
| | (23.03±2.98) | (26.90±2.03) | (**13.78**±0.94) | (11.59±1.98) | (18.82) | |
| Transformer+GIN +DA-W+SSReg | 77.57±1.14 | 79.44±1.21 | 65.27±1.49 | 62.28±1.71 | 71.14 | 3.6 |
| | (**23.13**±0.64) | (**28.97**±2.22) | (11.88±0.89) | (**13.24**±2.49) | (**19.30**) | |
| Transformer+GIN +DA-W+MFRReg | **77.63**±1.00 | **80.81**±1.27 | **68.56**±0.88 | **63.74**±0.27 | **72.68** | 1.2 |
| | (**23.83**±2.75) | (**29.04**±0.62) | (**13.94**±0.47) | (**16.80**±2.34) | (**20.90**) | |

**Table 2:** Unsupervised transfer of cross-species protein-protein physical interaction prediction.

| Methods | Physical: Node transfer ⇒ SSReg | | | | | |
| | Mouse | Zebrafish | Fruit fly | Yeast | Mean↑ | Rank↓ |
| --- | --- | --- | --- | --- | --- | --- |
| Mashup | 51.54±3.82 | 37.82±3.43 | 46.88±6.87 | 57.99±2.28 | 48.55 | 9.0 |
| | (5.58±0.35) | (3.98±0.12) | (7.19±3.93) | (6.78±0.92) | (5.88) | |
| D-SCRIPT | 58.22±6.97 | 49.58±1.12 | 62.97±0.78 | 62.43±0.59 | 58.30 | 8.0 |
| | (7.03±1.09) | (5.02±0.76) | (9.61±0.21) | (8.56±0.15) | (7.55) | |
| GraphCL | 76.88±0.42 | 79.11±1.14 | 81.02±0.98 | 71.03±0.30 | 77.01 | 6.0 |
| | (31.16±1.43) | (41.80±3.20) | (38.63±2.30) | (14.58±1.16) | (31.54) | |
| Transformer | 77.65±0.84 | 75.61±1.86 | 76.90±1.64 | 67.86±0.61 | 74.50 | 5.6 |
| | (**35.05**±0.92) | (**45.13**±3.15) | (32.72±2.34) | (12.46±1.08) | (31.34) | |
| Transformer +GIN | 79.77±0.92 | 80.85±2.41 | **82.38**±1.13 | 71.54±0.36 | 78.63 | 4.3 |
| | (31.23±1.94) | (34.29±12.42) | (**42.40**±2.04) | (15.73±0.79) | (30.91) | |
| Transformer +GIN+DA-C | 80.14±1.86 | **83.58**±1.15 | 81.49±1.27 | 71.30±0.61 | **79.12** | 3.3 |
| | (34.29±4.12) | (44.01±4.00) | (38.94±2.36) | (**16.80**±0.65) | (33.51) | |
| Transformer +GIN+DA-W | 80.18±1.38 | 80.88±3.08 | 81.51±0.36 | **72.66**±0.36 | 78.80 | 3.6 |
| | (34.14±0.85) | (41.88±2.15) | (42.02±0.69) | (16.18±2.67) | (**33.55**) | |
| Transformer+GIN +DA-W+SSReg | **81.20**±0.25 | 81.69±1.55 | **81.79**±0.74 | **73.07**±0.30 | **79.43** | 1.3 |
| | (**35.99**±1.51) | (**45.15**±2.07) | (**43.44**±1.16) | (**17.39**±1.01) | (**35.49**) | |
| Transformer+GIN +DA-W+MFRReg | **80.93**±1.11 | **81.95**±1.77 | 80.15±1.07 | 72.22±0.67 | 78.81 | 3.6 |
| | (34.63±3.71) | (43.09±4.19) | (35.43±1.60) | (16.40±1.12) | (32.38) | |

*co-expression* and experiments (we refer to it as *physical* to prevent confusion) channels (to be predicted) are experimental which deals with expensive functional genomics experiments (Parkinson et al., 2009) or direct lab assays (Brückner et al., 2009). More details of PPI data are shown in Appendix D, and ablations in Appendix G.

**Configurations.** We train models on PPIs of Homo Sapiens (human) which contain abundant PPI evidences (Ewing et al., 2007) and evaluate the models in four other species: Mus Musculus (mouse), Danio Rerio (zebrafish), Drosophila Melanogaster (fruit fly) and Saccharomyces Cerevisiae (yeast), which are repeated for three times for statistical significance. In evaluation, we follow (Cho et al., 2016) to perform negative sampling for experimental PPIs, to ensure that the known interactions compose only 5%. We adopt Transformer (Tay et al., 2020) (v.s. HRNN (Karimi et al., 2019; 2020a)) to embed protein sequences and then GIN (Xu et al., 2018) (v.s. GAT (Veličković et al., 2017)) to perform message passing, with comparisons in Appendix E. The compared state-of-the-art (SOTA) approaches are Mashup (Cho et al., 2016) and D-SCRIPT (Sledzieski et al., 2021) designed for PPI prediction as well as GraphCL (You et al., 2020a) for general graph (self-supervised) representation learning. See Appendix E for details.

**Hypotheses.** We hypothesize that co-expression interactions are more associated with links (than physical interactions do, **link transfer**), based upon the existing findings that two genes are poten-

tially similar in expression profiles if they are with similar promoter regions (might indicate locating in the neighborhood) (Park et al., 2002), involved in fusion events (Fernebro et al., 2006), or sharing co-occurrence patterns (Larmuseau et al., 2019). For physical interactions, we hypothesize that they rely more on node features (sequences, **node transfer**) considering the recent breakthrough that protein structure information (and ultimately physical interactions and functions) can be recovered from sequences with high accuracy (Jumper et al., 2021). We calculate the homophily ratios (Zhu et al., 2020a; Pei et al., 2020) of 3 out of 4 species in Figure 2 as numerical evidence to support the hypotheses (see Appendix F for the computing procedure, where zebrafish is excluded due to the overly sparse (<200) physical interactions as shown in Appendix D).

**Results.** The results of unsupervised transfer are shown in Table 1 and 2. We put semi-supervised results in Appendix G with the similar findings. Through comparing between: (**i**) methods w/o and w/ DA techniques, (**ii**) methods w/o and w/ spectral regularization, and (**iii**) our proposed methods and SOTA competitors, we have the following findings.

(**i**) **Vanilla DA in general provides benefits though occasionally degrades performance.** With the assistance of domain-invariant representation learning, GNNs generally lead to a better performance with exceptions, i.e., with Wasserstein distance guided DA (DA-W, as in optimization (3) (Redko et al., 2017; Shen et al., 2018)), 11 out of 16 metrics turn higher than w/o DA in the unsupervised setting, and so do 10 out of 16 in semi-supervised. The occasional performance degrade fits our analysis in Section 4.2, that the optimized source risk and distribution divergence of representations could lead to either improved or deteriorated performances without guarantee.

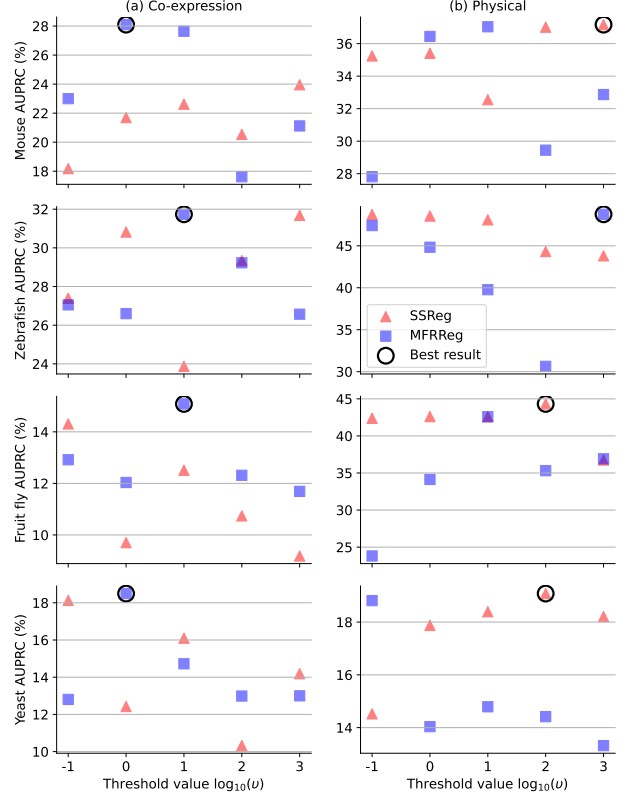

**Figure 3:** Spectral regularization performance of different threshold values $\upsilon$ in co-expression and physical interaction prediction (with trade-off factors $\gamma'$ selected via validation).

Similar phenomenon happens to DA with domain classifiers (DA-C, (Ganin et al., 2016)). Comparison between DA-W and DA-C shows that, DA-W performs better than DA-C when transferring to graphs with the larger domain gap (i.e. from human to fruit fly and yeast, justified by their biological taxonomic ranks and phylogenetic distances (Alberts et al., 2002), see Appendix D for details).

(**ii**) **In a principled way, spectral regularization further boosts GNN performance with DA-W and consistently alleviates performance degradation.** Under the regularization on GNN spectral properties, domain-invariant representation learning provides further benefits in a principled way. Specifically, MFRReg improves 13 out of 16 co-expression interaction prediction versus DA-W, since it assists GNN to mine from the computational PPIs that co-expression interactions are more correlated with, as hypothesized in Section 5.1. We show in Figure 3 that it is a consequence of a better sweet spot when regularizing MFR, which echos our analysis in Section 4.1, that MFRReg benefits more in the link transfer setting. Moreover, SSReg improves 13 out of 16 physical interaction prediction which reinforces GNN to dig more from protein sequence embeddings that physical interactions are more relevant to. This echos our analysis that SSReg benefits more in the node transfer setting, conforming to our theory-grounded regularization design.

Besides, we observe that the improvements are more significant in the unsupervised setting than semi-supervised. A possible reason is that, under the guidance of certain target labelled data during transferring, DA-W are less prone to capture superfluous information even without regularization.

(iii) **Integrating protein sequences together with computational PPIs leads to better transfer performance.** Comparing with SOTA competitors, we find that utilizing computational PPIs alone (such as Mashup) or utilizing protein sequences alone (such as D-SCRIPT which heavily relies on the pre-trained sequence encoder on a tremendous and diverse population of protein sequences or even structures (Bepler & Berger, 2019)) leads to less competitive results than integrating them together when transferring across species. For self-supervised pre-training methods (such as GraphCL), the existence of domain gap prompts "negative transfer" in the unsupervised setting, which is alleviated under the guidance of certain target labels in the semi-supervised setting.

## 5.2 CLASSIFYING PAPER TOPICS OF DIFFERENT TIME PERIODS

**Datasets and Configurations.** We also conduct experiments on the benchmark datasets provided in ArnetMiner (Tang et al., 2008), to classify paper topics in the temporally shifting citation networks. Papers published between 2000 and 2010 are collected from ACM and those after 2010 are from DBLP, with statistics shown in Appendix D. The Citation database is not examined here without processed data public. The networks are built with nodes for papers in six categories (to be predicted) and links representing citations. Following (Wu et al., 2020), we train methods on ACM/DBLP and evaluate them on DBLP/ACM, which are repeated for ten times. The compared SOTAs include graph representation learning w/o and w/ DA techniques. Our implementation is built

**Table 3:** Unsupervised transfer of paper topic classification in temporally evolved citation networks. Numbers in **red** are the best accuracies (mean±std%), which without standard deviation are from (Wu et al., 2020).

| Methods | Link transfer ⇒ MFRReg | |
|---|---|---|
| | ACM→DBLP | DBLP→ACM |
| DeepWalk | 17.98 | 36.49 |
| LINE | 19.72 | 41.17 |
| GraphSAGE | 72.28 | 69.61 |
| DNN | 42.79 | 59.04 |
| GCN | 64.86 | 69.45 |
| DGRL (Ganin et al., 2016) | 43.03 | 59.47 |
| AdaGCN (Sun et al., 2019) | 71.42 | 70.45 |
| EERM (Wu et al., 2022a) | 64.95±1.18 | 63.65±0.31 |
| UDAGCN (Wu et al., 2020) | 80.08±0.88(73.41) | 75.55±0.31(76.17) |
| GNN+DA-W | 89.08±0.10 | 75.82±0.05 |
| GNN+DA-W +SSReg | 90.90±0.08 | 76.15±0.06 |
| GNN+DA-W +MFRReg | **91.65**±0.06 | **76.26**±0.05 |

upon UDAGCN (Wu et al., 2020). See Appendix E for more details. We also assay our methods on the large-scale OGB benchmark (Hu et al., 2020).

**Hypothesis.** Paper topics are verified to have strong homophily with citations mostly (Zhu et al., 2020a; Pei et al., 2020) (**link transfer**), which is adopted as the hypothesis in our study.

**Results.** The results of unsupervised transfer are shown in Table 3, demonstrating the applicability of the proposed spectral regularization in different applications. Comparing graph representation learning w/o and w/ DA techniques, domain-invariant representations generally improve performances. Further

**Table 4:** Paper topic classification on ogbn-arxiv under different label rates. Reported numbers are accuracy (%).

| Label Rate | Link transfer ⇒ MFRReg | | |
|---|---|---|---|
| | 1% | 10% | 100% |
| GNN | 61.63±1.19 | 69.00±0.37 | 71.85±0.19 |
| GNN+DA-W | 63.96±0.61 | 68.98±0.28 | 71.93±0.16 |
| GNN+DA-W+SSReg | 63.53±0.89 | 69.18±0.31 | 71.82±0.28 |
| GNN+DA-W+MFRReg | **64.22**±0.51 | **69.21**±0.38 | **72.03**±0.23 |

built upon the SOTA UDAGCN, applying DA-W to minimize the domain divergence on representations benefits the transfer performance from ACM to DBLP while hurting it from DBLP to ACM, which is consistent with the observation (**i**). Similar results (also referring to Appendix G, Table 8 for the semi-supervised setting) on the large-scale ogbn-arxiv dataset are shown in Table 3.

Via spectral regularization, specifically MFRReg to exploit the invariant topology information across domains that is more related to node labels as hypothesized, our methods achieve better performance than all competitors, which is consistent with the observation (**ii**) in Section 5.1.

## 6 CONCLUSIONS

To fulfill the practical demands of transfer learning on graph data, we develop theory-grounded spectral regularization for GNNs to learn transferable graph representations. We first leverage domain adaptation with optimal transport theory to dive into the guaranteed bound for the transfer performance. This bound reveals that varying the Lipschitz constant of the GNNs could lead to a tighter bound by balancing domain divergence and GNN power. Building on the graph filter theory, we next show that one can regularize GNN spectral properties to regularize the Lipschitz constant, which motivates us to propose spectral regularizations. Numerical results conform to our theoretical analysis that regularizing SS and MFR brings benefits to the scenarios of node transfer and link transfer, respectively, in both the unsupervised and supervised settings.

ACKNOWLEDGEMENT

This project was in part supported by the National Institute of General Medical Sciences (R35GM124952 to Y.S.), the National Science Foundation (CCF-1943008 to Y.S.), and the US Army Research Office Young Investigator Award (W911NF2010240 to Z.W.). Portions of this research were conducted with the advanced computing resources provided by Texas A&M High Performance Research Computing.

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

APPENDIX

## A    PROOF FOR LEMMA 1

☐ **Lemma 1.** Suppose that $\mathcal{G}$ is the set for graphs of the size $N_G$ after padding with isolated nodes, similar to (Zhu et al., 2021). Given $\forall G_1, G_2 \in \mathcal{G}$ and $A_1 = U_1 \Lambda_1 U_1^\mathsf{T}, A_2 = U_2 \Lambda_2 U_2^\mathsf{T}$, the eigen decomposition for adjacency matrices $A_1$ and $A_2$ that $\Lambda_1 = \mathrm{diag}([\lambda_{1,1}, ..., \lambda_{1,N_G}]), \Lambda_2 = \mathrm{diag}([\lambda_{2,1}, ..., \lambda_{2,N_G}])$ (eigen values are sorted in the descending order). A GNN is constructed by composing a graph filter and nonlinear mapping that $f(G_1) = r(\sigma(\mathcal{S}(A_1)X_1 W)) = r(\sigma(U_1 \mathcal{S}(\Lambda_1) U_1^\mathsf{T} X_1 W))$ where $\mathcal{S}$ is the polynomial function that $\mathcal{S}(A_1) = \sum_{k=0}^{\infty} s_k A_1^k$, $W \in \mathcal{R}^{D \times D'}$ is the learnable weight matrix, $r$ is the mean/sum/max readout function to pool node representations, and the pointwise nonlinearity holds as $|\sigma(b) - \sigma(a)| \le |b - a|, \forall a, b \in \mathcal{R}$. Assuming $\|X\|_{\mathrm{op}} \le 1$ and $\|W\|_{\mathrm{op}} \le 1$, the following inequality holds:

$$\|f(G_1) - f(G_2)\|_2 \le C_\lambda (1 + \tau \sqrt{N_G}) \|A_1 - P^* A_2 P^{*\mathsf{T}}\|_\mathsf{F} + \mathcal{O}(\|A_1 - P^* A_2 P^{*\mathsf{T}}\|_\mathsf{F}^2)$$
$$+ \max\left\{ |\mathcal{S}(\Lambda_2)| \right\} \|X_1 - P^* X_2\|_\mathsf{F},$$

where $\tau = (\|U_1 - U_2\|_\mathsf{F} + 1)^2 - 1$ stands for the eigenvector misalignment which can be bounded, $P^* = \arg\min_{P \in \Pi} \{ \|X_1 - P X_2\|_\mathsf{F} + \|A_1 - P A_2 P^\mathsf{T}\|_\mathsf{F} \}$, $\Pi$ is the set of permutation matrices, $\mathcal{O}(\|A_1 - P^* A_2 P^{*\mathsf{T}}\|_\mathsf{F}^2)$ is the remainder term with bounded multipliers defined in (Gama et al., 2020), and $C_\lambda$ is the spectral Lipschitz constant that $\forall \lambda_i, \lambda_j, |\mathcal{S}(\lambda_i) - \mathcal{S}(\lambda_j)| \le C_\lambda |\lambda_i - \lambda_j|$.

*Proof.* Denote the optimal permutation matrix for $G_1, G_2$ as $P^*$, we compute the difference of the GNN outputs:

$$\|f(G_1) - f(G_2)\|_2$$
$$= \|r(\sigma(\mathcal{S}(A_1)X_1 W)) - r(\sigma(\mathcal{S}(A_2)X_2 W))\|_2$$
$$\overset{(a)}{=} \|r(\sigma(\mathcal{S}(A_1)X_1 W)) - r(\sigma(\mathcal{S}(P^* A_2 P^{*\mathsf{T}})P^* X_2 W))\|_2$$
$$\overset{(b)}{\le} \|\mathcal{S}(A_1)X_1 W - \mathcal{S}(P^* A_2 P^{*\mathsf{T}})P^* X_2 W\|_\mathsf{F}$$
$$\overset{(c)}{\le} \|W\|_{\mathsf{op}} \Big( \|\mathcal{S}(A_1)X_1 - \mathcal{S}(P^* A_2 P^{*\mathsf{T}})X_1 + \mathcal{S}(P^* A_2 P^{*\mathsf{T}})X_1 - \mathcal{S}(P^* A_2 P^{*\mathsf{T}})P^* X_2\|_\mathsf{F} \Big)$$
$$\overset{(d)}{\le} \|W\|_{\mathsf{op}} \|X_1\|_{\mathsf{op}} \|\mathcal{S}(A_1) - \mathcal{S}(P^* A_2 P^{*\mathsf{T}})\|_\mathsf{F} + \|W\|_{\mathsf{op}} \|\mathcal{S}(P^* A_2 P^{*\mathsf{T}})\|_{\mathsf{op}} \|X_1 - P^* X_2\|_\mathsf{F}$$
$$\overset{(e)}{\le} \|\mathcal{S}(A_1) - \mathcal{S}(P^* A_2 P^{*\mathsf{T}})\|_\mathsf{F} + \max(|\mathcal{S}(\Lambda_2)|) \|X_1 - P^* X_2\|_\mathsf{F}$$
$$\overset{(f)}{\le} C_\lambda (1 + \tau \sqrt{N_G}) \|A_1 - P^* A_2 P^{*\mathsf{T}}\|_\mathsf{F} + \mathcal{O}(\|A_1 - P^* A_2 P^{*\mathsf{T}}\|_\mathsf{F}^2) + \max(|\mathcal{S}(\Lambda_2)|) \|X_1 - P^* X_2\|_\mathsf{F},$$

where (a) is due to the permutation invariance property of graph filters; (b) is achieved with the triangle inequality and the assumption $|\sigma(b) - \sigma(a)| \le |b - a|, \forall a, b \in \mathcal{R}$; (c) and (d) use the fact that for any two matrices $A, B$, $\|AB\|_\mathsf{F} \le \min(\|A\|_{\mathsf{op}} \|B\|_\mathsf{F}, \|A\|_\mathsf{F} \|B\|_{\mathsf{op}})$, and (c) further applies the triangle inequality; (e) adopts the assumption $\|X\|_{\mathrm{op}} \le 1, \|W\|_{\mathrm{op}} \le 1$ which in practice can be guaranteed with normalization, and easily extended to the case with $\|X\|_{\mathrm{op}} \le K, \|W\|_{\mathrm{op}} \le K, \forall K > 0$, and because $\mathcal{S}(P^* A_2 P^{*\mathsf{T}}) = (P^* U_2) \mathcal{S}(\Lambda_2)(P^* U_2)^\mathsf{T}$ can be diagonalized, its operator norms equal the spectral radius; (f) is the direct outcome borrowed from (Gama et al., 2020) Theorem 1. We complete the proof.

## B    PROOF FOR LEMMA 2

☐ **Lemma 2.** Suppose that $\mathcal{G}$ is the set for graphs of the size $N_G$ after padding with isolated nodes, similar to (Zhu et al., 2021). Define the matching distance between $G_1, G_2 \in \mathcal{G}$ as $\eta(G_1, G_2) = \min_{P \in \Pi} \{ \|X_1 - P X_2\|_\mathsf{F} + \|A_1 - P A_2 P^\mathsf{T}\|_\mathsf{F} \}$. Suppose that the edge perturbation is bounded that $\forall G_1, G_2 \in \mathcal{G}, \|A_1 - P^* A_2 P^{*\mathsf{T}}\|_\mathsf{F} \le \varepsilon$ with the optimal permutation $P^*$, and there exists an eigenvalue $\lambda^* \in \mathcal{R}$ to achieve the maximum $|\mathcal{S}(\lambda^*)| < \infty$. We can then calculate the Lipschitz constant of GNN as:

$$C_f = \max\left\{ C_\lambda K_1 + \varepsilon K_2, |\mathcal{S}(\lambda^*)| \right\},$$

where $K_1, K_2$ is the supremes of $(1 + \tau\sqrt{N_G})$ and the remainder multiplier in Lemma 1, respectively, following similar philosophies in ((Gama et al., 2020), Theorem 1).

*Proof.* To calculate the Lipschitz constant $C_f$ w.r.t the matching distance, based upon Lemma 1, we assure the following inequality:

$$\|f(G_1) - f(G_2)\|_2 \leq C_\lambda(1 + \tau\sqrt{N_G})\|A_1 - P^*A_2P^{*\mathsf{T}}\|_\mathsf{F} + \mathcal{O}(\|A_1 - P^*A_2P^{*\mathsf{T}}\|_\mathsf{F}^2)$$
$$+ |\mathcal{S}(\lambda^*)|\|X_1 - P^*X_2\|_\mathsf{F},$$
$$\leq C_f\eta(G_1, G_2),$$

the latter inequality of which can be rewritten as:

$$\Big(C_\lambda(1 + \tau\sqrt{N_G})\|A_1 - P^*A_2P^{*\mathsf{T}}\|_\mathsf{F} + \mathcal{O}(\|A_1 - P^*A_2P^{*\mathsf{T}}\|_\mathsf{F}^2) - C_f\|A_1 - P^*A_2P^{*\mathsf{T}}\|_\mathsf{F}\Big)$$
$$+ (|\mathcal{S}(\lambda^*)| - C_f)\|X_1 - P^*X_2\|_\mathsf{F} \leq 0,$$

which is necessary for:

$$C_\lambda(1 + \tau\sqrt{N_G})\|A_1 - P^*A_2P^{*\mathsf{T}}\|_\mathsf{F} + \mathcal{O}(\|A_1 - P^*A_2P^{*\mathsf{T}}\|_\mathsf{F}^2) - C_f\|A_1 - P^*A_2P^{*\mathsf{T}}\|_\mathsf{F} \leq 0,$$
$$(|\mathcal{S}(\lambda^*)| - C_f)\|X_1 - P^*X_2\|_\mathsf{F} \leq 0,$$

which is equivalent to:

$$C_f \geq C_\lambda K_1 + \varepsilon K_2,$$
$$C_f \geq |\mathcal{S}(\lambda^*)|.$$

The bounding of $K_1, K_2$ follows (Gama et al., 2020) Theorem 1 and the first minimum solution can be calculated from the quadratic function w.r.t. the edge matching distance $\|A_1 - P^*A_2P^{*\mathsf{T}}\|_\mathsf{F}$. Let $C_f$ takes the larger value between them, we complete the proof.

## C    PROOF FOR LEMMA 3

☐ **Lemma 3.** Under the assumption of Corollary 1, we further assume that there exists a small amount of i.i.d. samples with labels $\{(G_n, Y_n)\}_{n=1}^{N_\mathrm{T}'}$ from the target distribution $\mathbb{P}_\mathrm{T}(G, Y)$ ($N_\mathrm{T}' \ll N_\mathrm{S}$) and bring in the conditional shift assumption that domains have different labeling function $\hat{h}_\mathrm{S} \neq \hat{h}_\mathrm{T}$ and $\max_{G_1, G_2} \frac{|\hat{h}_\mathrm{D}(G_1) - \hat{h}_\mathrm{D}(G_2)|}{\eta(G_1, G_2)} = C_h \leq C_f C_g$ ($\mathrm{D} \in \{\mathrm{S}, \mathrm{T}\}$) for some constant $C_h$ and distance measure $\eta$. Let $\mathcal{H} := \{h : \mathcal{G} \to \mathcal{Y}\}$ be the set of bounded real-valued functions with the pseudo-dimension $Pdim(\mathcal{H}) = d$, with probability at least $1 - \delta$ the following inequality holds:

$$\epsilon_\mathrm{T}(h, \hat{h}_\mathrm{T}) \leq \frac{N_\mathrm{T}'}{N_\mathrm{S} + N_\mathrm{T}'}\hat{\epsilon}_\mathrm{T}(h, \hat{h}_\mathrm{T}) + \frac{N_\mathrm{S}}{N_\mathrm{S} + N_\mathrm{T}'}\hat{\epsilon}_\mathrm{S}(h, \hat{h}_\mathrm{S})$$
$$+ \frac{N_\mathrm{S}}{N_\mathrm{S} + N_\mathrm{T}'}\Big(2C_f C_g W_1(\mathbb{P}_\mathrm{S}(G), \mathbb{P}_\mathrm{T}(G)) + \omega$$
$$+ [\frac{8d}{N_\mathrm{T}'}\log(\frac{eN_\mathrm{T}'}{d}) + \frac{2}{N_\mathrm{T}'}\log(\frac{1}{\delta}) + \frac{8d}{N_\mathrm{S}}\log(\frac{eN_\mathrm{S}}{d}) + \frac{2}{N_\mathrm{S}}\log(\frac{1}{\delta})]^{\frac{1}{2}}\Big),$$

where $\omega = \min\big(|\epsilon_\mathrm{S}(h, \hat{h}_\mathrm{S}) - \epsilon_\mathrm{S}(h, \hat{h}_\mathrm{T})|, |\epsilon_\mathrm{T}(h, \hat{h}_\mathrm{S}) - \epsilon_\mathrm{T}(h, \hat{h}_\mathrm{T})|\big)$.

*Proof.* Before showing the designated lemma, we first introduce the following inequality to be used that:

$$|\epsilon_\mathrm{S}(h, \hat{h}_\mathrm{S}) - \epsilon_\mathrm{T}(h, \hat{h}_\mathrm{T})|$$
$$= |\epsilon_\mathrm{S}(h, \hat{h}_\mathrm{S}) - \epsilon_\mathrm{S}(h, \hat{h}_\mathrm{T}) + \epsilon_\mathrm{S}(h, \hat{h}_\mathrm{T}) - \epsilon_\mathrm{T}(h, \hat{h}_\mathrm{T})|$$
$$\leq |\epsilon_\mathrm{S}(h, \hat{h}_\mathrm{S}) - \epsilon_\mathrm{S}(h, \hat{h}_\mathrm{T})| + |\epsilon_\mathrm{S}(h, \hat{h}_\mathrm{T}) - \epsilon_\mathrm{T}(h, \hat{h}_\mathrm{T})|$$
$$\overset{(a)}{\leq} |\epsilon_\mathrm{S}(h, \hat{h}_\mathrm{S}) - \epsilon_\mathrm{S}(h, \hat{h}_\mathrm{T})| + 2C_f C_g W_1\big(\mathbb{P}_\mathrm{S}(G), \mathbb{P}_\mathrm{T}(G)\big),$$

where (a) results from (Shen et al., 2018) Lemma 1 with the assumption $\max(\|h\|_{\mathrm{Lip}}, \max_{G_1, G_2} \frac{|\hat{h}_\mathrm{D}(G_1) - \hat{h}_\mathrm{D}(G_2)|}{\eta(G_1, G_2)}) \leq C_f C_g, \mathrm{D} \in \{\mathrm{S}, \mathrm{T}\}$. Similarly, we obtain:

$$|\epsilon_\mathrm{S}(h, \hat{h}_\mathrm{S}) - \epsilon_\mathrm{T}(h, \hat{h}_\mathrm{T})| \leq |\epsilon_\mathrm{T}(h, \hat{h}_\mathrm{S}) - \epsilon_\mathrm{T}(h, \hat{h}_\mathrm{T})| + 2C_f C_g W_1\big(\mathbb{P}_\mathrm{S}(G), \mathbb{P}_\mathrm{T}(G)\big).$$

We therefore combine them into:

$$|\epsilon_S(h, \hat{h}_S) - \epsilon_T(h, \hat{h}_T)| \leq 2C_f C_g W_1\Big(\mathbb{P}_S(G), \mathbb{P}_T(G)\Big) + \min\Big(|\epsilon_S(h, \hat{h}_S) - \epsilon_S(h, \hat{h}_T)|, |\epsilon_T(h, \hat{h}_S) - \epsilon_T(h, \hat{h}_T)|\Big),$$

i.e. the following holds to bound the target risk $\epsilon_T(h, \hat{h}_T)$:

$$\epsilon_T(h, \hat{h}_T) \leq \epsilon_S(h, \hat{h}_S) + 2C_f C_g W_1\Big(\mathbb{P}_S(G), \mathbb{P}_T(G)\Big) + \min\Big(|\epsilon_S(h, \hat{h}_S) - \epsilon_S(h, \hat{h}_T)|, |\epsilon_T(h, \hat{h}_S) - \epsilon_T(h, \hat{h}_T)|\Big).$$

We next link the bound with the empirical risk and labeled sample size by showing, with probability at least $1 - \delta$ that:

$$\epsilon_T(h, \hat{h}_T) \leq \epsilon_S(h, \hat{h}_S) + 2C_f C_g W_1\Big(\mathbb{P}_S(G), \mathbb{P}_T(G)\Big) + \min\Big(|\epsilon_S(h, \hat{h}_S) - \epsilon_S(h, \hat{h}_T)|, |\epsilon_T(h, \hat{h}_S) - \epsilon_T(h, \hat{h}_T)|\Big)$$

$$\leq \hat{\epsilon}_S(h, \hat{h}_S) + 2C_f C_g W_1\Big(\mathbb{P}_S(G), \mathbb{P}_T(G)\Big) + \min\Big(|\epsilon_S(h, \hat{h}_S) - \epsilon_S(h, \hat{h}_T)|, |\epsilon_T(h, \hat{h}_S) - \epsilon_T(h, \hat{h}_T)|\Big)$$
$$+ \sqrt{\frac{2d}{N_S} \log(\frac{eN_S}{d})} + \sqrt{\frac{1}{2N_S} \log(\frac{1}{\delta})},$$

and:

$$\epsilon_T(h, \hat{h}_T) \leq \hat{\epsilon}_T(h, \hat{h}_T) + \sqrt{\frac{2d}{N'_T} \log(\frac{eN'_T}{d})} + \sqrt{\frac{1}{2N'_T} \log(\frac{1}{\delta})},$$

which results from (Mohri et al., 2018) Theorem 11.8. Lastly, we combine the above two inequalities, with probability at least $1 - \delta$ that:

$$\epsilon_T(h, \hat{h}_T)$$

$$\overset{(a)}{\leq} \frac{N'_T}{N_S + N'_T}\Big(\hat{\epsilon}_T(h, \hat{h}_T) + \sqrt{\frac{2d}{N'_T} \log(\frac{eN'_T}{d})} + \sqrt{\frac{1}{2N'_T} \log(\frac{1}{\delta})}\Big)$$
$$+ \frac{N_S}{N_S + N'_T}\Big(\hat{\epsilon}_S(h, \hat{h}_S) + \sqrt{\frac{2d}{N_S} \log(\frac{eN_S}{d})} + \sqrt{\frac{1}{2N_S} \log(\frac{1}{\delta})}\Big)$$
$$+ \frac{N_S}{N_S + N'_T}\Big(2C_f C_g W_1(\mathbb{P}_S(G), \mathbb{P}_T(G)) + \min(|\epsilon_S(h, \hat{h}_S) - \epsilon_S(h, \hat{h}_T)|, |\epsilon_T(h, \hat{h}_S) - \epsilon_T(h, \hat{h}_T)|)\Big)$$

$$\overset{(b)}{\leq} \frac{N'_T}{N_S + N'_T}\Big(\hat{\epsilon}_T(h, \hat{h}_T) + \sqrt{\frac{4d}{N'_T} \log(\frac{eN'_T}{d}) + \frac{1}{N'_T} \log(\frac{1}{\delta})}\Big)$$
$$+ \frac{N_S}{N_S + N'_T}\Big(\hat{\epsilon}_S(h, \hat{h}_S) + \sqrt{\frac{4d}{N_S} \log(\frac{eN_S}{d}) + \frac{1}{N_S} \log(\frac{1}{\delta})}\Big)$$
$$+ \frac{N_S}{N_S + N'_T}\Big(2C_f C_g W_1(\mathbb{P}_S(G), \mathbb{P}_T(G)) + \min(|\epsilon_S(h, \hat{h}_S) - \epsilon_S(h, \hat{h}_T)|, |\epsilon_T(h, \hat{h}_S) - \epsilon_T(h, \hat{h}_T)|)\Big)$$

$$\overset{(c)}{\leq} \frac{N'_T}{N_S + N'_T}\hat{\epsilon}_T(h, \hat{h}_T) + \frac{N_S}{N_S + N'_T}\hat{\epsilon}_S(h, \hat{h}_S)$$
$$+ \frac{N_S}{N_S + N'_T}\Big(2C_f C_g W_1(\mathbb{P}_S(G), \mathbb{P}_T(G)) + \min(|\epsilon_S(h, \hat{h}_S) - \epsilon_S(h, \hat{h}_T)|, |\epsilon_T(h, \hat{h}_S) - \epsilon_T(h, \hat{h}_T)|)$$
$$+ [\frac{8d}{N'_T} \log(\frac{eN'_T}{d}) + \frac{2}{N'_T} \log(\frac{1}{\delta}) + \frac{8d}{N_S} \log(\frac{eN_S}{d}) + \frac{2}{N_S} \log(\frac{1}{\delta})]^{\frac{1}{2}}\Big),$$

where (a) is the outcome of applying the union bound with coefficient $\frac{N'_T}{N_S + N'_T}, \frac{N_S}{N_S + N'_T}$ respectively; (b) and (c) result from the Cauchy-Schwartz inequality and (c) additionally adopt the assumption $N'_T \ll N_S$, following the sleight-of-hand in (Li et al., 2021a) Theorem 3.2.

## D   DATASET STATISTICS

For PPI data, we construct the graph with nodes as protein sequences (using protein language modeling (Tay et al., 2020; Karimi et al., 2019) for node features), and edges as computational interactions

of *neighborhood*, *fusion* and *co-occurrence* channels (Szklarczyk et al., 2021) (i.e. the edge feature dimension is 3).

Dataset statistics for PPI and citation networks are shown in Figure 5 and 6, respectively. For the species involved in PPI networks, we depict their relationship in biological taxonomic ranks in Figure 4, showing the domain gap between species in concept.

**Table 5:** Dataset statistics of PPI networks of different species. # denotes "number of".

| Species | # Node | # Edge | # Co-expression | # Physical |
|---|---|---|---|---|
| Human | 8,369 | 201,164 | 15,344 | 68,812 |
| Mouse | 55,55 | 55,668 | 4,490 | 11,264 |
| Zebrafish | 4,625 | 60,077 | 16,044 | 150 |
| Fruit fly | 5,503 | 108,858 | 25,404 | 14,694 |
| Yeast | 4,286 | 207,641 | 39,040 | 60,716 |

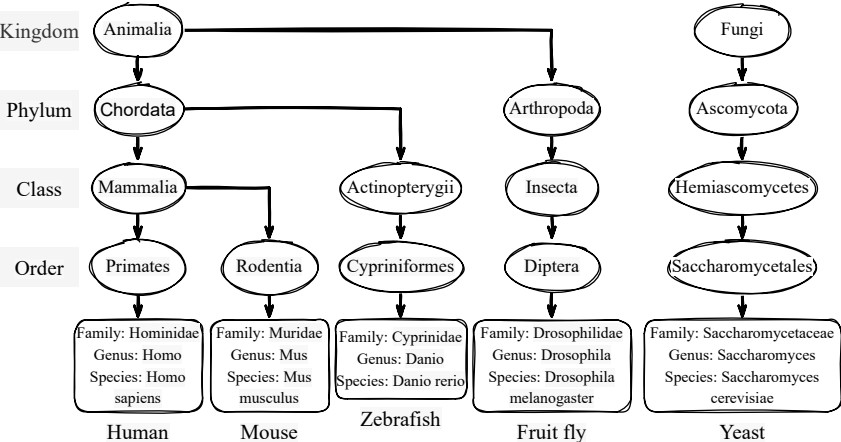

**Figure 4:** The relationship among species in biological taxonomic ranks.

**Table 6:** Dataset statistics of citation networks from different sources (in different time periods). # denotes "number of".

| Sources (periods) | # Node | # Edge | # Feature | # Class |
|---|---|---|---|---|
| DBLP (2000–2010) | 5,578 | 7,341 | 7,537 | 6 |
| ACM (2010– ) | 7,410 | 11,135 | 7,537 | 6 |

# E    DETAILED CONFIGURATIONS

The assayed data are released under the MIT license, and to our best knowledge, contain no privacy-infringing contents. Experiments are distributed on computer clusters with Tesla K80 GPU (11 GB memory) and NVIDIA A100 GPU (40 GB memory).

## E.1    WASSERSTEIN-1 DISTANCE ESTIMATION

We follow the most routine procedure (Shen et al., 2018) to estimate Wasserstein-1 distance of graph representations between distributions in adversarial training (3). Specifically, given the encoder $f$

and source and target data distributions $\mathbb{P}_S(G), \mathbb{P}_T(G)$, we estimate the distance as:

$$\hat{W}_1\Big(\mathbb{P}_S(G), \mathbb{P}_T(G)\Big) = \max_{\|\hat{f}\| \leq 1} \mathbb{E}_{\mathbb{P}_S(G)} \hat{f}(f(G)) - \mathbb{E}_{\mathbb{P}_T(G)} \hat{f}(f(G)),$$

where the critic function $\hat{f}$ is instantiated with a multilayer perceptron and satisfies $\|\hat{f}\| = \sup \frac{\|\hat{f}(x) - \hat{f}(y)\|_2}{\|x-y\|_2} \leq 1$, which is achieved by enforcing the gradient penalty $(\|\nabla_x \hat{f}(x)\|_2 - 1)^2$. The final estimation is thus implemented as:

$$\hat{W}_1\Big(\mathbb{P}_S(G), \mathbb{P}_T(G)\Big) = \max_{\hat{f}} \mathbb{E}_{\mathbb{P}_S(G)} \hat{f}(f(G)) - \mathbb{E}_{\mathbb{P}_T(G)} \hat{f}(f(G)) - \hat{\gamma}(\|\nabla_x \hat{f}(x)\|_2 - 1)^2,$$

where we follow (Shen et al., 2018; Gulrajani et al., 2017) to set $\hat{\gamma} = 10$ by default.

### E.2 PREDICTING PROTEIN-PROTEIN INTERACTIONS ACROSS VARIOUS SPECIES

During collecting PPI data from the STRING database (Szklarczyk et al., 2021), we abandon the interactions in channels of *neighborhood_transferred*, *coexpression_transferred*, and *experiments_transferred* to prevent information leakage across species (especially from supervisions of co-expression and physical interaction prediction). For co-expression and physical interactions, we use the high-quality threshold of 0.7 (Szklarczyk et al., 2021) to convert them into binary labels.

We use Sinkhorn Transformer (Tay et al., 2020) (a variant of Transformer with sparse attention) with depth 4, attention head 4 and bucket size 32 to embed protein sequences, and further adopt GIN (Xu et al., 2018) with depth 3 and MLP depth 2 to perform message passing across proteins. We also try HRNN (Karimi et al., 2019; 2020a) with k-mer 75 to embed protein sequence and GAT (Veličković et al., 2017) with depth 3 and attention head 8 to perform message passing, with comparison on human to yeast transfer shown in Table 7, which states in our case that, (Sinkhorn) Transformer and GIN outperform HRNN and GAT, respectively. In training, we hold out 20% of human PPIs for validation. For the semi-supervised setting, 0.1% of experimental PPIs are in addition available during training. We train with convergence assured for 500 epochs with learning rate 0.0001, hidden dimension 256 and batch size 128 which is sampled by random walk, optimized by Adam optimizer. For domain-invariant representation learning guided by Wasserstein distance (as described in optimization (3)) please refer to (Shen et al., 2018) for implementation, and see (Ganin et al., 2016) for one with domain classifiers. We selected the trade-off factor $\gamma$ in optimization (3) from {1e-1, 1e-2, 1e-3, 1e-4, 1e-5, 1e-6} through validation. In evaluation, we follow (Cho et al., 2016) to perform negative sampling for experimental PPIs, to ensure that the known interactions compose only 5%, with the assumption the number of unobserved interactions is only a small fraction.

**Table 7:** Comparisons between different protein sequence encoders and GNNs in the human to yeast transfer setting.

| Methods | Co-expression | | Physical | |
|---|---|---|---|---|
| | AUORC (%) | AUPRC (%) | AUORC (%) | AUPRC (%) |
| HRNN | 61.55±1.55 | 8.91±1.37 | 65.24±1.17 | 9.71±0.86 |
| Transformer | 60.66±1.11 | 9.11±1.42 | 66.65±2.28 | 11.75±1.46 |
| Transformer+GAT | 60.85±1.98 | 9.00±1.55 | 70.51±1.31 | 14.53±2.27 |
| Transformer+GIN | 61.18±2.63 | 10.90±1.40 | 71.15±1.44 | 15.02±1.79 |

On baseline implementation, for Mashup (Cho et al., 2016), we apply the official MATLAB code to extract protein representation w.r.t. computational PPIs in the neighborhood, fusion and co-occurrence channels, on top of which we apply a two-layer MLP to train with labels; for D-SCRIPT (Sledzieski et al., 2021), we apply the official PyTorch APIs to perform the sequence embedding projection, inter-protein residue contact and interaction modeling, where we use the same Sinkhorn Transformer as in our methods to generate sequence embeddings rather than the large model pre-trained on additional data of protein sequences with structures, for fair comparison and also complexity consideration; for GraphCL (You et al., 2020a), we follow the main idea and implement it with the node masking augmentation, where we randomly replace 20% of animo acids in each protein sequences with mask tokens.

### E.3 CLASSIFYING PAPER TOPICS OF DIFFERENT TIME PERIODS

We follow the same experiment setting as in UDAGCN (Wu et al., 2020), which we implement DA-W and spectral regularization on. The compared SOTAs without DA techniques are DeepWalk (Perozzi et al., 2014), LINE (Tang et al., 2015), GraphSAGE (Hamilton et al., 2017), DNN (MLP on node features alone), and GCN (Kipf & Welling, 2016), and ones with DA techniques are DGRL (Ganin et al., 2016), AdaGCN (Dai et al., 2019), and UDAGCN (Wu et al., 2020). We notice that the reported results in (Wu et al., 2020) are not directly comparable with (Mao et al., 2021) due to different experimental configurations (e.g. validation partition). By applying DA-W, we replace the domain classifier with the Wasserstein distance critic (Shen et al., 2018) to learn domain-invariant representations, and further implement our proposed spectral regularization.

## F HOMOPHILY RATIO COMPUTATION PROCEDURE

Given a graph $G = \{V, E\}$ with the set of nodes $V$ and edges $E$ as described in Section 2. Denote the labeled edge set as $E'$, to quantify the association between $E$ and $E'$, we borrow the idea from (Zhu et al., 2020a; Pei et al., 2020) to calculate the homophily ratio as:

$$\text{Hom} = \frac{1}{|V|} \sum_{v \in V} \frac{|\{(v, w) : w \in \mathcal{N}(v) \text{ and } (v, w) \in E'\}|}{|\mathcal{N}(v)|},$$

where $\mathcal{N}(v)$ is the set of neighbor nodes of $v$ determined by $E$. In cross-species PPI prediction, $E'$ is the experimental (co-expression or physical) PPIs. To measure the association between computational and experimental PPIs, we define $E$ as the set of edges if there is any computational PPI (neighborhood, fusion or co-occurrence) value greater than the medium threshold of 0.4 (Szklarczyk et al., 2021). To measure the association between protein sequences and experimental PPIs, we first calculate the sequence identity (Karimi et al., 2020b) between all protein pairs, and define $E$ as the set of edges if the sequence identity is greater than 0.3 (Pearson, 2013).

## G MORE EXPERIMENTAL RESULTS

Results of semi-supervised cross-species protein-protein interaction prediction are shown in Tables 8, 9; semi-supervised paper topic classification in Table 10; ablation studies of model architectures and adversarial training strategies in Tables 11, 12.

**Table 8:** Semi-supervised transfer of cross-species protein-protein co-expression interaction prediction. Numbers in red are the top-2 AUROC (AUPRC) (mean±std%). DA-C, DA-W denotes domain-invariant representation learning with domain classifiers (Ganin et al., 2016) and guided by Wasserstein distance as in optimization (3) (Redko et al., 2017; Shen et al., 2018), respectively.

| Methods | Co-expression: Link transfer ⇒ MFRReg | | | | | |
| | Mouse | Zebrafish | Fruit fly | Yeast | Mean↑ | Rank↓ |
|---|---|---|---|---|---|---|
| Mashup | 49.06±6.33 (5.66±0.73) | 51.80±5.64 (5.64±0.61) | 53.86±2.04 (5.86±0.27) | 48.79±5.36 (5.18±0.64) | 50.87 (5.58) | 9.0 |
| D-SCRIPT | 52.85±0.45 (6.07±0.10) | 61.76±3.45 (11.12±1.94) | 61.01±1.14 (8.59±0.20) | 56.52±0.71 (7.64±0.15) | 58.03 (8.35) | 7.7 |
| GraphCL | 74.62±2.75 (17.54±4.70) | 75.26±0.64 (22.93±2.42) | 64.55±1.70 (11.26±0.98) | **70.15**±0.97 (**20.75**±3.84) | 71.14 (18.12) | 4.8 |
| Transformer | 71.57±1.51 (20.39±0.90) | 69.87±0.52 (26.09±1.58) | 61.76±2.71 (12.14±1.60) | 67.01±0.70 (14.11±1.24) | 67.55 (18.18) | 6.2 |
| Transformer +GIN | **77.86**±3.13 (**22.17**±7.86) | **80.39**±3.79 (**30.15**±5.26) | 66.92±1.65 (8.04±0.53) | 67.07±0.71 (14.43±1.92) | 73.06 (18.69) | 3.5 |
| Transformer +GIN+DA-C | 75.13±3.22 (17.71±1.39) | 79.01±0.96 (28.54±1.88) | 66.02±1.19 (8.34±0.41) | 68.12±0.96 (17.65±1.69) | 72.07 (18.06) | 4.8 |
| Transformer +GIN+DA-W | 77.06±0.74 (20.63±1.81) | 79.48±1.39 (28.96±2.75) | **71.11**±2.11 (**16.04**±0.94) | 67.60±2.78 (15.14±2.09) | **73.81** (**20.19**) | **3.0** |
| Transformer+GIN +DA-W+SSReg | 76.65±1.44 (20.66±4.17) | **80.19**±0.74 (27.23±0.75) | **69.36**±2.25 (**13.56**±2.54) | 67.58±2.56 (18.46±2.22) | **73.44** (19.97) | 3.2 |
| Transformer+GIN +DA-W+MFRReg | **77.26**±0.91 (**24.16**±0.95) | 79.19±1.24 (**30.02**±0.29) | 66.58±1.24 (12.33±0.73) | **68.90**±1.13 (**20.10**±3.28) | 72.98 (**21.65**) | **2.5** |

**Table 9:** Semi-supervised transfer of cross-species protein-protein physical interaction prediction.

| Methods | Physical: Node transfer ⇒ SSReg | | | | | |
| --- | --- | --- | --- | --- | --- | --- |
| | Mouse | Zebrafish | Fruit fly | Yeast | Mean↑ | Rank↓ |
| Mashup | 57.90±3.07 (9.23±4.18) | 63.88±7.27 (7.63±1.36) | 46.84±6.61 (5.93±1.32) | 51.68±5.48 (5.89±1.22) | 55.07 (7.17) | 8.7 |
| D-SCRIPT | 49.34±0.67 (5.87±0.09) | 68.61±3.47 (19.36±4.99) | 69.79±1.79 (20.25±0.78) | 59.66±1.02 (7.46±0.15) | 61.85 (13.23) | 8.2 |
| GraphCL | 79.08±0.63 (34.96±1.31) | 82.61±1.41 (**49.81**±2.91) | 81.04±0.62 (38.65±1.40) | 70.06±3.10 (13.33±3.69) | 78.19 (34.18) | 4.6 |
| Transformer | 77.44±0.84 (34.63±0.77) | 77.50±1.20 (**46.49**±2.33) | 78.14±0.29 (34.56±1.54) | 69.15±1.00 (14.64±1.88) | 75.55 (32.58) | 6.0 |
| Transformer +GIN | 79.76±2.26 (34.58±1.90) | 81.09±1.46 (41.46±4.02) | **82.41**±0.96 (40.14±5.24) | 72.59±1.08 (16.76±1.45) | 78.96 (33.23) | 4.1 |
| Transformer +GIN+DA-C | **80.47**±0.60 (**35.62**±0.43) | 79.88±1.01 (42.28±2.92) | 81.09±1.02 (37.37±1.59) | 71.42±0.67 (15.62±2.01) | 78.21 (32.72) | 4.5 |
| Transformer +GIN+DA-W | 79.72±1.13 (35.47±2.79) | **83.45**±3.44 (45.05±2.37) | **82.36**±1.21 (**44.07**±3.42) | **73.33**±1.84 (18.78±5.12) | **79.71** (**35.84**) | **2.6** |
| Transformer+GIN +DA-W+SSReg | **80.69**±0.40 (**36.40**±1.34) | **84.69**±2.29 (45.93±3.64) | 81.92±0.81 (**41.73**±2.55) | **73.01**±0.98 (**19.08**±1.63) | **80.07** (**35.78**) | **1.8** |
| Transformer+GIN +DA-W+MFRReg | 79.73±0.86 (31.98±1.54) | 81.69±1.08 (42.36±0.38) | 79.54±1.20 (38.97±1.57) | 72.64±0.85 (**19.42**±1.29) | 78.40 (33.18) | 4.2 |

**Table 10:** Paper topic semi-supervised classification on ogbn-arxiv under different label rates. Reported numbers are accuracy (%).

| Label Rate | Link transfer ⇒ MFRReg | | |
| --- | --- | --- | --- |
| | 1% | 10% | 100% |
| GNN | 66.64±0.64 | 69.35±0.52 | 71.96±0.29 |
| GNN+DA-W | 67.53±0.58 | 68.82±0.43 | 71.98±0.21 |
| GNN+DA-W+SSReg | 67.71±0.43 | 69.92±0.28 | 72.04±0.17 |
| GNN+DA-W+MFRReg | **67.73**±0.42 | **69.99**±0.37 | **72.06**±0.29 |

**Table 11:** Unsupervised transfer of cross-species (human to yeast) protein-protein co-expression interaction prediction. Numbers in **red** are the best performance among the sub-row.

| Methods | Co-expression: Link transfer ⇒ MFRReg | | Physical: Node transfer ⇒ SSReg | |
| --- | --- | --- | --- | --- |
| | AUROC (%) | AUPRC (%) | AUROC (%) | AUPRC (%) |
| Transformer+GIN | **63.91**±1.55 | 11.15±1.11 | 71.54±0.36 | 15.73±0.79 |
| Transformer+GIN+SSReg | 61.63±0.41 | 13.62±1.79 | **71.89**±0.78 | **17.80**±1.58 |
| Transformer+GIN+MFRReg | 63.01±3.05 | **15.32**±1.84 | 71.83±1.17 | 17.01±2.09 |
| Transformer+GIN+DA-C | 60.65±3.85 | 10.72±2.44 | 71.30±0.61 | 16.80±0.65 |
| Transformer+GIN+DA-C+SSReg | 61.28±2.79 | 12.39±3.33 | **73.07**±0.85 | **18.13**±2.39 |
| Transformer+GIN+DA-C+MFRReg | **66.03**±0.85 | **14.51**±2.90 | 71.61±0.70 | 16.01±1.42 |

**Table 12:** Unsupervised transfer of cross-species (human to fruit fly) protein-protein co-expression interaction prediction..

| Methods | Co-expression: Link transfer ⇒ MFRReg | | Physical: Node transfer ⇒ SSReg | |
| --- | --- | --- | --- | --- |
| | AUROC (%) | AUPRC (%) | AUROC (%) | AUPRC (%) |
| Transformer+GIN | **66.54**±1.11 | 13.48±0.71 | 82.38±1.13 | **42.40**±2.04 |
| Transformer+GIN+SSReg | 63.12±0.77 | 8.91±1.44 | **82.50**±0.54 | 40.90±2.92 |
| Transformer+GIN+MFRReg | 66.18±1.94 | **14.14**±2.41 | 81.76±0.71 | 38.15±2.79 |
| Transformer+GIN+DA-C | 64.78±1.23 | 11.61±2.08 | 81.49±1.27 | 38.94±2.36 |
| Transformer+GIN+DA-C+SSReg | 66.30±1.51 | **13.10**±1.41 | **82.32**±0.70 | 38.87±1.82 |
| Transformer+GIN+DA-C+MFRReg | **66.92**±1.26 | 11.45±2.55 | 82.28±0.68 | **41.29**±1.76 |

# H  COMPLEXITY ANALYSIS OF SPECTRAL REGULARIZATION

**Time.** Let's consider a graph of $N$ nodes. The spectral regularization is composed of two steps: extracting spectral signals (with eigenvalue decomposition (EVD) involved, complexity $\mathcal{O}(N^3)$)

and neural network propagation (complexity $\mathcal{O}(N)$ (You et al., 2020c)). Thus, in theory, the former step dominates the time complexity.

In practice, when training on large networks, mini-batch training is commonly adopted (Hamilton et al., 2017; Zeng et al., 2019) to avoid computational burden and empirically for better generalizability (Hamilton et al., 2017), where the node number in each batch is restricted to be much smaller than the whole network. In our experiments, batch size is set as 128, with $\leq 3$ seconds cost brought for each epoch (see Table 13), which is five times less than the cost brought by adversarial training (20 seconds, see Table 13). We also provide the time consumption for the full training in Table 15, including the best-performed SOTA GraphCL, where the time consumption for GraphCL was around 2 times higher in pretraining and 2 times lower in finetuning compared with training from scratch. Nevertheless, training from scratch did not take longer than 10 hours. We further provide Table 14 to demonstrate the numerical running time of EVD is acceptable as we see in Table 14, with batch size $\leq 2048$.

Nevertheless, if it is believed that there are indispensable benefits from the full-batch training, we can always perform the full-batch EVD for only once before training, for the repetitive usage later.

Moreover, we would like to clarify that the spectral properties of graph filters in theory do not have to depend on inputs (sampled graphs or not, an analog is in signal processing, a low-pass system, here "low-pass" is the spectral property, would suppress high frequencies for any input signal).

In details, suppose that the graph filter function $S(\cdot)$ with a certain domain $D_1$ (which is a polynomial function resided in a GNN, see Lemma 1) is constructed (or regularized) to be $C_\lambda$-Lipschitz and bounded by $B$ (i.e. $\forall x \in D, |S(x)| \leq B$), then for an arbitrary graph with the adjacency matrix $A = U\Lambda U^{\mathsf{T}}$ with all eigenvalues range in $D_2$ (i.e. $\forall \lambda \in \lambda, \lambda \in D_2$), the following applies. If $D_2 \subseteq D_1$, the graph filter $S(A) = US(\Lambda)U^{\mathsf{T}}$ (Ortega et al., 2018) preserves the properties of (i) spectral smoothness (SS) with $C_\lambda$ that $\forall \lambda_i, \lambda_j, \frac{|S(\lambda_i) - S(\lambda_j)|}{|\lambda_i - \lambda_j|} \leq C_\lambda$, and (ii) maximum frequency response (MFR) $B$ that $\max |S(\lambda)| \leq B$. We can see from the above statement that, spectral properties of SS and MFR are formulated on eigenvalues but not eigenvectors.

In practice, the spectral regularization is performed in the spectra of sampled networks (with eigenvalues in $\Lambda_B$ range in $D_B$) and the readers might concern whether the regularized properties in $D_B$ still hold in the spectra of ego-graphs of nodes (with eigenvalues in $\Lambda_E$ range in $D_E$) since whether $D_E \subseteq D_B$ is unclear. According to the eigenvalue interlacing theorem (Haemers, 1995), since adjacency matrices of ego-graphs are principal submatrices of sampled networks, we have $\min(\Lambda_B) \leq \min(\Lambda_E) \leq \max(\Lambda_E) \leq \max(\Lambda_B)$ that essentially leads to $D_E \subseteq D_B$. Therefore, regularized spectral properties on spectra $D_E$ (in practice) should be preserved on spectra $D_B$ (in theory).

**Memory.** Beyond time complexity, memory consumption results from data processing (while all compared methods use the same amount of data) and model propagation. Compared to the best-performed SOTA GraphCL, memory consumption was similar between GraphCL and Transformer+GIN(+DA-W+SpecReg), since GraphCL also uses the same Transformer+GIN backbone architecture to extract protein representations (Transformer to embed protein sequences as vertex features, and then GIN to conduct message passing along topology, please refer to Appendix E for details). GPU memory taken by Transformer+GIN was around 24GB which is within the capacity of the conventional NVIDIA A100 GPU (40 GB memory).

The improving AUPRC of PPI prediction with regard to such affordable computational resources is significant. The computational resource is usually not the bottleneck for in-silico methods (in our case, an NVIDIA A100 GPU + less than 10 hours), while detecting PPI (usually highly imbalanced Rao et al. (2014)) via wet laboratories is very costly with expensive reagents for weeks or months. For instance, the routine yeast two-hybrid system requires steps including building vectors, transforming plasmids, cell culture, luciferase assay, etc Brückner et al. (2009), where accurate predictions could play a critical role in accelerating the process.

## I    LEMMA 4: LIPSCHITZ CONSTANT OF TWO-LAYER GNN

$\square$ **Lemma 4.** Suppose that $\mathcal{G}$ is the set for graphs of the size the size $N_G$ after padding with isolated nodes, similar to (Zhu et al., 2021). Following the setting in Lemma 1 and 2, a GNN layer is

**Table 13:** Running time for one epoch training in unsupervised cross-species (human to yeast) protein-protein interaction prediction.

| | Transformer +GIN | Transformer +GIN+DA-W | Transformer+GIN +DA-W+SSReg | Transformer+GIN +DA-W+MFRReg |
|---|---|---|---|---|
| Time | 45s | 65s | 68s | 66s |

**Table 14:** Running time for eigenvalue decomposition under different batch sizes in unsupervised cross-species (human to yeast) protein-protein interaction prediction.

| Batch size | 32 | 128 | 512 | 2048 |
|---|---|---|---|---|
| Time | 0.005s | 0.01s | 0.08s | 3.63s |

**Table 15:** Running time for full training in unsupervised cross-species (human to yeast) protein-protein interaction prediction.

| | GraphCL Pretraining | GraphCL Finetuning | GraphCL Pretraining+Finetuning | Transformer +GIN | Transformer +GIN+DA-W | Transformer+GIN +DA-W+SSReg | Transformer+GIN +DA-W+MFRReg |
|---|---|---|---|---|---|---|---|
| Time | 12.22h | 3.75h | 15.97h | 6.25h | 9.02h | 9.44h | 9.16h |

constructed as $f^{(\cdot)}(G) = \sigma(\mathcal{S}^{(\cdot)}(A)XW^{(\cdot)})$, and a two-layer GNN as $f(G) = r \circ f^{(2)} \circ f^{(1)}(G)$, where $r$ is the mean/sum/max readout function to pool node representations. Denote the SS and MFR terms of $f^{(l)}$ as $t_{\text{SS}}^{(l)} = C_\lambda^{(l)}K_1 + \varepsilon K_2, t_{\text{MFR}}^{(l)} = |\mathcal{S}^{(l)}(\lambda^*)|$ that $C_f^{(l)} = \max\{t_{\text{SS}}^{(l)}, t_{\text{MFR}}^{(l)}\}, l \in \{1, 2\}$ per Lemma 2, we can then calculate the Lipschitz constant of GNN as:

$$C_f = \max\left\{t_{\text{SS}}^{(2)} + t_{\text{MFR}}^{(2)}t_{\text{SS}}^{(1)}, t_{\text{MFR}}^{(2)}t_{\text{MFR}}^{(1)}\right\}.$$

*Proof.* The key step is to recognize that the input of the 2nd GNN layer is actually a graph with the same adjacency matrix $A$, but a different node feature matrix $X^{(1)} = f^{(1)}(G)$. Denote the optimal permutation matrix for $G_1, G_2$ as $P^*$, we thus compute the difference of the GNN outputs:

$$\|f(G_1) - f(G_2)\|_2$$

$$= \|(r \circ f^{(2)}) \circ f^{(1)}(G_1) - (r \circ f^{(2)}) \circ f^{(1)}(G_2)\|_2$$

$$\overset{(a)}{\leq} C_\lambda^{(2)}(1 + \tau\sqrt{N_G})\|A_1 - P^*A_2P^{*\mathsf{T}}\|_{\mathsf{F}} + \mathcal{O}(\|A_1 - P^*A_2P^{*\mathsf{T}}\|_{\mathsf{F}}^2) + \max(|\mathcal{S}^{(2)}(\Lambda_2)|)\|X_1^{(1)} - P^*X_2^{(1)}\|_{\mathsf{F}}$$

$$\overset{(b)}{\leq} C_\lambda^{(2)}(1 + \tau\sqrt{N_G})\|A_1 - P^*A_2P^{*\mathsf{T}}\|_{\mathsf{F}} + \mathcal{O}(\|A_1 - P^*A_2P^{*\mathsf{T}}\|_{\mathsf{F}}^2)$$

$$+ \max(|\mathcal{S}^{(2)}(\Lambda_2)|)\Big(C_\lambda^{(1)}(1 + \tau\sqrt{N_G})\|A_1 - P^*A_2P^{*\mathsf{T}}\|_{\mathsf{F}} + \mathcal{O}(\|A_1 - P^*A_2P^{*\mathsf{T}}\|_{\mathsf{F}}^2)$$

$$+ \max(|\mathcal{S}^{(1)}(\Lambda_2)|)\|X_1 - P^*X_2\|_{\mathsf{F}}\Big)$$

$$= \Big(C_\lambda^{(2)}(1 + \tau\sqrt{N_G}) + \max(|\mathcal{S}^{(2)}(\Lambda_2)|)C_\lambda^{(1)}(1 + \tau\sqrt{N_G})\Big)\|A_1 - P^*A_2P^{*\mathsf{T}}\|_{\mathsf{F}}$$

$$+ \Big(1 + \max(|\mathcal{S}^{(2)}(\Lambda_2)|)\Big)\mathcal{O}(\|A_1 - P^*A_2P^{*\mathsf{T}}\|_{\mathsf{F}}^2) + \max(|\mathcal{S}^{(2)}(\Lambda_2)|)\max(|\mathcal{S}^{(1)}(\Lambda_2)|)\|X_1 - P^*X_2\|_{\mathsf{F}},$$

where (a), (b) are due to the reuse the inequalities (a)-(f) in Lemma 1. Next, following the same spirit in Lemma 2, to calculate the Lipschitz constant of $f$, we assure the inequality:

$$\Big(C_\lambda^{(2)}(1 + \tau\sqrt{N_G}) + |\mathcal{S}^{(2)}(\lambda^*)|C_\lambda^{(1)}(1 + \tau\sqrt{N_G})\Big)\|A_1 - P^*A_2P^{*\mathsf{T}}\|_{\mathsf{F}}$$

$$+ \Big(1 + |\mathcal{S}^{(2)}(\lambda^*)|\Big)\mathcal{O}(\|A_1 - P^*A_2P^{*\mathsf{T}}\|_{\mathsf{F}}^2) + |\mathcal{S}^{(2)}(\lambda^*)\mathcal{S}^{(1)}(\lambda^*)|\|X_1 - P^*X_2\|_{\mathsf{F}} \leq C_f\eta(G_1, G_2),$$

that is:

$$\Big(C_\lambda^{(2)}(1 + \tau\sqrt{N_G}) + |\mathcal{S}^{(2)}(\lambda^*)|C_\lambda^{(1)}(1 + \tau\sqrt{N_G}) - C_f\Big)\|A_1 - P^*A_2P^{*\mathsf{T}}\|_{\mathsf{F}}$$

$$+ \Big(1 + |\mathcal{S}^{(2)}(\lambda^*)|\Big)\mathcal{O}(\|A_1 - P^*A_2P^{*\mathsf{T}}\|_{\mathsf{F}}^2) + \Big(|\mathcal{S}^{(2)}(\lambda^*)\mathcal{S}^{(1)}(\lambda^*)| - C_f\Big)\|X_1 - P^*X_2\|_{\mathsf{F}} \leq 0,$$

which is necessary for:

$$\left(C_\lambda^{(2)}(1 + \tau\sqrt{N_G}) + |\mathcal{S}^{(2)}(\lambda^*)|C_\lambda^{(1)}(1 + \tau\sqrt{N_G}) - C_f\right)\|A_1 - P^*A_2 P^{*\mathsf{T}}\|_\mathsf{F}$$
$$+ \left(1 + |\mathcal{S}^{(2)}(\lambda^*)|\right)\mathcal{O}(\|A_1 - P^*A_2 P^{*\mathsf{T}}\|_\mathsf{F}^2) \leq 0,$$
$$\left(|\mathcal{S}^{(2)}(\lambda^*)\mathcal{S}^{(1)}(\lambda^*)| - C_f\right)\|X_1 - P^*X_2\|_\mathsf{F} \leq 0,$$

which is equivalent to:

$$C_f \geq (C_\lambda^{(2)}K_1 + \varepsilon K_2) + |\mathcal{S}^{(2)}(\lambda^*)|(C_\lambda^{(1)}K_1 + \varepsilon K_2),$$
$$C_f \geq |\mathcal{S}^{(2)}(\lambda^*)\mathcal{S}^{(1)}(\lambda^*)|.$$

