# OpenReview forum: "Graph Domain Adaptation via Theory-Grounded Spectral Regularization"
_ICLR.cc/2023/Conference — ICLR 2023 poster_

### Official Review · Reviewer_xYtM · 2022-10-23

**Confidence:** 3
**Correctness:** 4
**Technical Novelty And Significance:** 3
**Empirical Novelty And Significance:** 3
**Recommendation:** 6

**Clarity, Quality, Novelty And Reproducibility:**

The paper is clearly written, and the readability is high. Clear recognition of the node transfer and link transfer is significant, though I'm not sure the concept is new. The regularization tailored for each scenario is an interesting insight in this work.

**Strength And Weaknesses:**

Strength
- In the paper, the transfer learning on the graph is roughly categorized into node transfer and link transfer. For each setting of the transfer learning, an appropriate regularization for the feature extractor is presented. An important insight provided by this paper is to recognize two types of information transformations on the graph. For each type, a reasonable regularization is proposed.

Weaknesses
- Only a single-layer architecture is used as the feature extractor f(G). Extension to multi-layer networks may be rather straightforward, though computational cost becomes high. A brief comment on such an extension can be beneficial for readers.
- The categorization of node transfer and link transfer is interesting and important. Usually, we may not have sufficient prior knowledge of the scenario. Is it possible to determine which scenario occurs in practice? Which kind of information is beneficial to identify which scenario occurs?


**Summary Of The Paper:**

The paper studies learning with graphs under the covariate shift assumption. The predictor is decomposed into two functions; one is the feature extractor of the graph, f, and the other is the discriminator, g. The generalization performance of the predictor h(G)=g(f(G)) for a graph G is governed by the Lipschitz constant of f and g. Through analysis of the theoretical properties of the feature extractor defined on the set of graphs, the authors proposed two regularization methods; spectral smoothness(SS) for the node transfer and maximum frequency response(MFR) for the link transfer. The theoretical findings well explain numerical results for link prediction and node classification.

**Summary Of The Review:**

The paper is clearly written. The regularization methods for node transfer and link transfer are an interesting contribution of this paper.

---

> ### Author Response · Authors · 2022-11-14
> **Response to Reviewer xYtM**
>
> **Q.** Only a single-layer architecture is used as the feature extractor f(G). Extension to multi-layer networks may be rather straightforward, though computational cost becomes high. A brief comment on such an extension can be beneficial for readers.
>
> **A.** We thank the reviewer for the interest in our work. Actually our theory is not limited to the single-layer GNN and we will communicate this better in the revision. We have additionally provided theoretical results for a two-layer GNN case in Appendix I that can be easily extended to multiple layers.
>
> ---------------------------------------------------------
>
> **Q.** The categorization of node transfer and link transfer is interesting and important. Usually, we may not have sufficient prior knowledge of the scenario. Is it possible to determine which scenario occurs in practice? Which kind of information is beneficial to identify which scenario occurs?
>
> **A.** The reviewer asked a very important question. Our answer is affirmative. In practice when domain knowledge is missing, we could use preliminary labeled data to calculate the node and link homophily ratios (as we did in our PPI experiments to support the hypothesis in Section 5.1, see Figure 2 and Appendix F) to suggest the potential transfer scenario.

---

### Official Review · Reviewer_bQEk · 2022-10-24

**Confidence:** 3
**Correctness:** 3
**Technical Novelty And Significance:** 3
**Empirical Novelty And Significance:** 3
**Recommendation:** 6

**Clarity, Quality, Novelty And Reproducibility:**

- The paper is well-written and easy to follow.
- Although the basis theorem is from the existing work of generic domain adaptation, the theory-driven regularization terms are impressive.
- The details of experiments are provided and the codes are submitted as supplementary.

**Strength And Weaknesses:**

### Strength
- The paper is well-written and the idea is well-motivated.
- The proposed theory-driven regularization terms are reliable and interesting. To be honest, I enjoy reading this (kind of) paper.
- The paper is technically sound.

### Weakness
- The theoretical analysis is limited to the single-layer GNN and it seems non-trivial to extend it into the multi-layer cases.
- The core theorem is mainly from the existing work of generic domain adaptation. The real theoretical contributions of this paper seem to be the two lemmas about the discussions of Lipschitz on graphs.


**Summary Of The Paper:**

This paper proposes theory-driven graph domain adaptation regularization terms, including spectral smoothing (SS) and maximum frequency response (MFR). The authors try to extend the theorem in generic domain adaptation into the graph field and focus on the definition of Lipschitz on GNN. Via analyzing the graph Lipschitz property, the authors develop two reguarlizations that directly minimize the upper-bound of the target risk (the right hand of the inequality). The experiments also verify the effectiveness of the idea.

**Summary Of The Review:**

I'd like to increase my score during the discussion period.

---

> ### Author Response · Authors · 2022-11-14
> **Response to Reviewer bQEk**
>
> **Q.** The theoretical analysis is limited to the single-layer GNN and it seems non-trivial to extend it into the multi-layer cases.
>
> **A.** We thank the reviewer for the comment. Actually, our theory is not limited to the single-layer GNN and we will communicate this better in the revision. We have additionally provided theoretical results for a two-layer GNN in Appendix I that can be easily extended to multiple layers.
>
> ----------------------------------------------------------------
>
> **Q.** The core theorem is mainly from the existing work of generic domain adaptation. The real theoretical contributions of this paper seem to be the two lemmas about the discussions of Lipschitz on graphs.
>
> **A.** We appreciate that the reviewer appropriately noted our contributions. We would like to take the opportunity to explain why these contributions are not as simple as they may appear. As we stated in the introduction (Section 1) and the method (Section 4), our theoretical results are indeed founded on the general DA theory yet instantiated for graph data. The instantiation part is nontrivial due to the discrete nature of graphs. More importantly, we connected our theory to two significant transfer scenarios of graphs in practice: node transfer and link transfer. Such connections and our corresponding contributions are unique for graphs and not present in the generic DA research.

---

> > ### Comment · Reviewer_bQEk · 2022-12-13
> > **Thanks for your response and sorry for the late feedback**
> >
> > I thank the authors for the kind response.  I actually read it weeks ago and keep reading other reviews. I appreciate the further analysis of multi-layer networks. The most impressive part for me is the discussions about graph Lipschitz. Overall, I give a positive score.

---

> > > ### Author Response · Authors · 2022-12-13
> > > **Thank You for Responses**
> > >
> > > Thank you for your thoughtful discussions and acknowledgment!

---

### Official Review · Reviewer_no2m · 2022-10-24

**Confidence:** 3
**Correctness:** 3
**Technical Novelty And Significance:** 2
**Empirical Novelty And Significance:** 3
**Recommendation:** 5

**Clarity, Quality, Novelty And Reproducibility:**

Based on the previous comments on Strength and Weakness, the article has only marginal novelties, with a strongt limitation of only working on the regularization w.r.t the Lipschitz constant.

The reproducibility is fair, as a code is provided (I didn't have the time to try it), yet not really good because the numerical cost is heavy. The authors mentioned for instance in SI, section E, the hardware used which appear to be a lot  with "computer clusters" in unprecised numbers.

The clarity and quality and faire as well, as I think that the experimental observations lack more robust insights and conclusions.

**Strength And Weaknesses:**

Strength:

1- the idea of adapting  [Redko et al., 2017]  to graphs is relevant ;

2- the obtained theoretical result (Corollary 1 and the two Lemmas of 4.1) appear to be sound.

Weaknesses:

1- the theoretical  results are obtained in a straightfoward re-purposing of the aforementioned works ; they don't look wrong, but not that original in lights of existing works.

2- the reach of this results is limited because, as soon as it is obtained, the authors limit themselves to the $C_f$ Lipschitz constant and the proposed method appear then to be only a slight improvement on existing architectures ;

3- the proposed regularizations, SSReg or MFRReg have an impact which is almost minor in Tables 1 and 2 ; and again in Tables 3 and 4 when comparing to GNN with DA-W ;

4- most of the really novel content is of experimental nature (in Section 5) and the observations reported in 5.1 and 5.2 appear to be quite fragile: i) I am not convinced by the paragraph "Hypotheses" in 5.1 on the two tasks on PPI and I think that more experimental validations should be sought for before concluding that ; and ii) the discussion under "Results" in 5.1 is not fully convincing about the impact of DA. Ifw we follow the metrics in term of ranks proposed by the authors, methods with DA-W appear to perform better in Tables 1 and 2 than without. Looking at specific instances is toio much cherry-picking of results that seem to saturate (the improvement by the proposed method is often on the third digit in percentage)

5- the article leverages many "big" models (Transformes, GNN with many layers) yet there is no major study about the computation power / time / expense incurred when doing this study (the only reported times are in Tables 13 and 14 in SI); The obtained marginal improvements should be discussed and put in balance with the increased computational costs. For instance, is the less than 5% increase in performance w.r.t. GraphCL  useful ? What's the difference in price (or resource consumption) between what is proposed and this latter method, for instance.


**Summary Of The Paper:**

The main idea of the article is to adapt the results about bounds in domain adaptation (from [Redko et al., 2017] and following wowrk) to situations involving graphs instead of Euclidean space. After a theoretical analysis, very close to the existing results in non-graph settings, the authors focus only on one term which drives the Lipschitz property of GNN (acting as a feature extractor) on graphs. Their methodological work then amounts to add a regularization over the graph spectrum based on spectral smoothness and/or maximum frequency response of the graph filter inside the GNN. Some numerical experiments on two types of datasets (Protein-Protein Interactions and Topic classification of articles) to study the possible performance of the method.

**Summary Of The Review:**

As a summary, given the aforementioned weaknesses, I currently vote against the acceptation of the article, which appears to me to be incremental both on the theoretical aspect and on the methodological+experimental side.

Update: scores raised after the answers.

---

> ### Author Response · Authors · 2022-11-14
> **Response to Reviewer no2m (1)**
>
> **Q.** Originality and usefulness of the theoretical results.
>
> **Q.1.** The theoretical results are obtained in a straightfoward re-purposing of the aforementioned works ; they don't look wrong, but not that original in lights of existing works.
>
> **A.** We thank the reviewer for the comment and would like to take the opportunity to argue that the contributions are not as straightforward as they may appear. Our theory is indeed founded on the general DA framework, as we stated in the introduction (Section 1) and the method (Section 4), yet with the following original advances:
>
> (i) We bridge our theory to two important transfer scenarios of graphs in practice: node transfer and link transfer (Section 4.2), via analyzing domain-divergence and discriminability terms w.r.t. graph labeling information. The connections provided guidance in selecting regularization a priori for experiments.  The way that we made such connections is unique for graphs and not present in the generic domain adaptation (DA) research or other graph DA works (see below).
>
> (ii) To form the basis for (i), we instantiated the general DA theory for graphs by leveraging the graph filter theory, through associating the GNN Lipschitz constant with two spectral properties (Section 4.1). The association is new to our knowledge, enabling the feasibility to modulate GNN Lipschitz for graph DA.
>
> **Q.2.** The reach of this results is limited because, as soon as it is obtained, the authors limit themselves to the Cf Lipschitz constant and the proposed method appear then to be only a slight improvement on existing architectures.
>
> **A.** We thank the reviewer for the concern but feel that this could be a misunderstanding.  We will do a better job in presenting the results during the revision.  And here we would like to clarify three points first.
>
> (i) The potential reach of the graph DA theory is more than Lipschitz. For instance, one can also reduce the domain-divergence term (W1 in Corollary 1) via designing appropriate graph data transformations/augmentations (e.g. efforts similar to [1,2] though their theoretical analysis is lacking).
>
> (ii) This paper focuses on modulating GNN Lipschitz to balance between domain-divergence and discriminability, motivated by the fact that Lipschitz is shown to have significant impacts on model behaviors, such as generalizability [3], robustness [4], transferability [5], and generation [6]. Accordingly, we propose GNN Lipschitz regularization for these known impacts.   Meanwhile, it can be relatively easy and effective to modulate GNN Lipschitz: for instance, one can carefully engineer GNN model architectures [7].
>
> (iii) Our future roadmap indeed includes the further development of the current graph DA theory (e.g. data transformations in (i)) and also the more unified pipeline to numerically instantiate the theoretical results.  We appreciate the reviewer’s encouragement to broaden the reach of our results.
>
> [1] Empowering Graph Representation Learning with Test-Time Graph Transformation
> [2] G-Mixup: Graph Data Augmentation for Graph Classification
> [3] Lipschitz regularity of deep neural networks: analysis and efficient estimation
> [4] Training robust neural networks using Lipschitz bounds
> [5] Wasserstein distance guided representation learning for domain adaptation
> [6] Improved training of wasserstein gans
> [7] Deep neural networks with trainable activations and controlled Lipschitz constant

---

> > ### Author Response · Authors · 2022-11-14
> > **Response to Reviewer no2m (2)**
> >
> > **Q.** Marginal numerical improvement.
> > The proposed regularizations, SSReg or MFRReg have an impact which is almost minor in Tables 1 and 2 ; and again in Tables 3 and 4 when comparing to GNN with DA-W.
> >
> > **A.** We thank the reviewer for the comment.  We are afraid that it could originate from some misunderstanding of our presentation.  While pledging to improve the presentation further, we would like to point out the following.
> >
> > (i) We respectfully argue that, in most cases, the numerical gains of spectral regularizations were not minor *within their designated transfer scenarios* (node/link transfer).
> >
> > - In PPI prediction, against SOTA methods, SSReg boosted the average AUROC/AUPRC(%) for its supposed transfer scenario - node transfer - by 0.63/1.94 (Table 2) and MFRReg did for its supposed transfer scenario - link transfer - by 0.98/2.08 (Table 1), where AUPRCis the more important metric for PPI due to the highly imbalanced labels.
> >
> > - In paper topic prediction, MFRReg boosted the average accuracy (%) by 1.5 (Table 3) and 0.26 (1% labeling rate in Table 4). The latter gain, seemingly small, might result from the largest dataset size of ogbn-arxiv (100 times larger than others), where adversarial training between source and target (DA-W) could have already provided sufficient benefits. However in practice, scenarios with limited data are ubiquitous and in great demand across many applications.
> >
> > (ii) More importantly, our theories are predictive and explanatory (Section 1 contribution bullet iii, Interpretation on how theory guides practice), as supported by empirical evidence, on which regularizer would gain more in which graph transfer scenario (Section 4.2).
> >
> > (iii) Lastly, we emphasize the consistency, that our observed numerical benefits are consistent across different experiments, reflected in the average rank boosting, e.g. by 2.4 in Table 1 and 2.0 in Table 2.
> >
> > ----------------------------------
> >
> > **Q.** Validity of experimental observations.
> > Most of the really novel content is of experimental nature (in Section 5) and the observations reported in 5.1 and 5.2 appear to be quite fragile:
> >
> > **Q.1.** I am not convinced by the paragraph "Hypotheses" in 5.1 on the two tasks on PPI and I think that more experimental validations should be sought for before concluding that ; and
> >
> > **A.** We provide numerical evidence for hypotheses in Figure 2 and Appendix F, via calculating homophily ratios on sequence identity and topology, to measure whether label information is preserved more in node features or edges. We add the highlight sentence after hypotheses.
> >
> > **Q.2.** the discussion under "Results" in 5.1 is not fully convincing about the impact of DA. Ifw we follow the metrics in term of ranks proposed by the authors, methods with DA-W appear to perform better in Tables 1 and 2 than without. Looking at specific instances is toio much cherry-picking of results that seem to saturate (the improvement by the proposed method is often on the third digit in percentage).
> >
> > **A.** We thank the reviewer for the comment. We meant to state in this observation that vanilla DA in general provides benefits while occasionally degrades performance, and our proposed regularization could provide more consistent improvement reflected in the average rank (e.g. boosting by 2.4 in Table 1 and 2.0 in Table 2). We revise the observation following your opinions.
> >
> > For the comment on the improvement, in most cases, the numerical gains of spectral regularizations were not minor *within their designated transfer scenarios* (node/link transfer). For instance, for PPI prediction, against SOTA methods, SSReg boosted the average AUROC/AUPRC(%) for its supposed transfer scenario - node transfer - by 0.63/1.94 (Table 2) and MFRReg did for its supposed transfer scenario - link transfer - by 0.98/2.08 (Table 1), where AUPRC is the more important metric for PPI due to the highly imbalanced labels.

---

> > > ### Author Response · Authors · 2022-11-14
> > > **Response to Reviewer no2m (3)**
> > >
> > > **Q.** Complexity of the proposed methods.
> > > The article leverages many "big" models (Transformes, GNN with many layers) yet there is no major study about the computation power / time / expense incurred when doing this study (the only reported times are in Tables 13 and 14 in SI); The obtained marginal improvements should be discussed and put in balance with the increased computational costs. For instance, is the less than 5% increase in performance w.r.t. GraphCL useful ? What's the difference in price (or resource consumption) between what is proposed and this latter method, for instance.
> > >
> > > **A.** We thank the reviewer for suggestions, following which we decompose the computational resource consumption (power/expense) into time and GPU memory. Memory consumption results from data processing (while all compared methods use the same amount of data) and model propagation. Compared to the best-performed SOTA GraphCL:
> > >
> > > (i) Memory consumption was similar between GraphCL and Transformer+GIN(+DA-W+SpecReg), since GraphCL also uses the same Transformer+GIN backbone architecture to extract protein representations (Transformer to embed protein sequences as vertex features, and then GIN to conduct message passing along topology, please refer to Appendix E for details). GPU memory taken by Transformer+GIN was around 24GB which is within the capacity of the conventional NVIDIA A100 GPU (40 GB memory).
> > >
> > > (ii) The time consumption for GraphCL was around 2 times higher in pretraining and 2 times lower in finetuning compared with training from scratch, as reported in Appendix H. Nevertheless, training from scratch did not take longer than 10 hours.
> > >
> > > (iii) Improving AUPRC of PPI prediction is useful. The computational resource is usually not the bottleneck for in-silico methods (in our case, an NVIDIA A100 GPU + less than 10 hours), while detecting PPI (usually highly imbalanced [1]) via wet laboratories is very costly with expensive reagents for weeks or months. For instance, the routine yeast two-hybrid system requires steps including building vectors, transforming plasmids, cell culture, luciferase assay, etc [2], where accurate predictions could play a critical role to accelerate the process.
> > >
> > > [1] Protein-protein interaction detection: methods and analysis
> > > [2] Yeast two-hybrid, a powerful tool for systems biology

---

> > > > ### Comment · Reviewer_no2m · 2022-11-18
> > > > **Acknowledging the answers**
> > > >
> > > > I have read the different reviews and all the answers provided in addition to the revisions done. It's possible that I indeed misundertood some aspects in the original presentation and the results apppear more solid now that when initially reviewing the article. I am not fully convinced by the answers to Q1 and Q2, about the originality w.r.t works in DA of Redko et al., and about the focus on $C_f$. Yet, I agree that, at least technically, the conversion of the results to two transfer scenarios is carefully done and that there is some value in that. Also, I understand better the authors' motivation to control the  GNN Lipschitz.
> > > >
> > > > With all that, I revising my notation to a better score.

---

> > > > ### Author Response · Authors · 2022-11-18
> > > > **Thank You for Responses**
> > > >
> > > > We genuinely appreciate the reviewer's informative responses to our answers and the acknowledgment of our work’s motivation and value.
> > > >
> > > > For the partially convinced Q1 and Q2 on theoretical contributions, we are happy to further discussions to address any remaining concrete concerns. A brief summary of our previous answers is as follows (for reference):
> > > >
> > > > - Generic DA (works in DA of Redko et al.) is not directly applicable to graphs;
> > > > - Our development enables one feasibility for such application (detailed in answers of Q1, originality);
> > > > - Such development needs not to be restricted to Lipschitz (detailed in answers of Q2, potential reaches).
> > > >
> > > > Any detailed critique would be greatly appreciated.

---

### Official Review · Reviewer_1Hvj · 2022-10-25

**Confidence:** 4
**Correctness:** 4
**Technical Novelty And Significance:** 3
**Empirical Novelty And Significance:** 3
**Recommendation:** 6

**Clarity, Quality, Novelty And Reproducibility:**

The paper is well written and easy to follow. Various technical steps taken by the authors are described with the help of examples and figures. The results in the paper are motivated by optimal-transport theory and the authors leverage the existing results on stability of graph filters to get around the challenges offered in the graph domain.

I would recommend that the authors provide additional details on the empirical evaluation of the Wasserstein distance $\hat W_1$ and the construction of graphs in the protein-protein interaction networks.

**Strength And Weaknesses:**

Strengths:
Domain adaptation is a very practically relevant problem. The proposed framework is theoretically motivated and provides a solid contribution in this regard. Experiments are well-designed and provide a convincing evidence for the approach.

Weaknesses:
I don't identify any major weaknesses. However, I couldn't verify the 'padding with isolated nodes' procedure. For $A_1$ in Lemma 1 (and other results where this procedure is mentioned), is this similar to extending the matrix diagonally with dummy nodes to make it of size $N_G \times N_G$? If so, this does not appear to be an optimal procedure for domain adaptation.



**Summary Of The Paper:**

This paper provides a framework for domain adaptation in graph neural networks. Algorithmically, the framework relies on addition of two regularizer terms to the loss function. The construction of the regularizer terms is motivated from theoretical results derived using the theory of optimal transport-based domain adaptation and stability of graph filters from the existing literature. Experiments on real world datasets illustrate the utility of the proposed framework.

**Summary Of The Review:**

I think this is a solid contribution in domain adaptation on graphs. The algorithms are well-motivated from theory and the experiments are illuminating and convincing.

---

> ### Author Response · Authors · 2022-11-14
> **Response to Reviewer 1Hvj**
>
> **Q.** I couldn't verify the 'padding with isolated nodes' procedure. For A1 in Lemma 1 (and other results where this procedure is mentioned), is this similar to extending the matrix diagonally with dummy nodes to make it of size NG×NG? If so, this does not appear to be an optimal procedure for domain adaptation.
>
> **A.** We thank the reviewer for comments and confirm that the reviewer understood the padding procedure correctly. In GNN theoretical studies with transferability or generalizability, it is a widely-adopted procedure [1,2] to handle graphs of varied sizes; and in practice, no padding is actually needed if the standard summation pooling [3] is adopted as in our experiments (Section 5, since isolated nodes have no contribution to graph representations), where numerical results were shown well-calibrated with theoretical analysis (contributions bullets iii, iv).
>
> [1] Transfer learning of graph neural networks with ego-graph information maximization, NeurIPS 2021
> [2] Stability and generalization of graph convolutional neural networks, KDD 2019
> [3] How powerful are graph neural networks? ICLR 2018
>
> ---------------------------------------------------
>
> **Q.** I would recommend that the authors provide additional details on the empirical evaluation of the Wasserstein distance W^1 and the construction of graphs in the protein-protein interaction networks.
>
> **A.** We thank the reviewer for the suggestion. We have now included additional details about estimating the first Wasserstein distance in Appendix E and those about the PPI graph construction in Appendix D.

---

### Author Response · Authors · 2022-11-17
**Gentle Reminder**

Dear reviewers,

Thank you all again for your first-round comments that have helped our revision. We hope that you have found our responses useful and our revision satisfactory. We would be thrilled to have more such exciting and constructive discussions with you. Could you please kindly share any comments to our responses at your earliest convenience so that we could still respond before the author-reviewer discussion window closes on November 18? Or if you have already found our responses satisfactory, we humbly remind you of a fitting update of the rating. Thank you all again for your time and efforts!

Sincerely, All authors

---

### Decision · Program_Chairs · 2023-01-20

**Decision:**

Accept: poster

**Justification For Why Not Higher Score:**

There is a concern (no2m) on minor performance improvement in experiments.


**Justification For Why Not Lower Score:**

The paper is well written and the proposal is supported by sound theoretical analysis. Risk bound on graph domain is also relative less common compared to ones on Euclidean domain. I think the paper is worth publishing.

**Metareview: Summary, Strengths And Weaknesses:**

The paper tackles the problem of covariate shifts when the input domain is graphs. Through analysis of the theoretical properties of the feature extractor defined on the set of graphs, the paper proposes two regularization methods; spectral smoothness(SS) for the node transfer and maximum frequency response(MFR) for the link transfer.

All reviewers agree that the paper is well written and the proposal is supported by sound theoretical analysis. The problem is well dissected and studied in the sense that transfer learning on graphs is categorized into node transfer and link transfer. For each setting of the transfer learning, an appropriate regularization for the feature extractor is presented (xYtM).

Regarding weaknesses, one concern raised was that Corollary 1 (risk bound) is a direct application of existing results (no2m, bQEk). Separately another concern was that theoretical analysis of graph neural networks is only presented for the single-layer case. Both of these concerns have been satisfactorily addressed by a revision and an appropriate rebuttal. A new analysis was added to the appendix for the two-layer case. Although there is a concern (no2m) on minor performance improvement in experiments, the overall theoretical contributions appear to be novel and outweigh this concern.

I recommend that the paper be accepted.

*Suggestions to the authors*:
Please address all comments from the reviewers. Please add a discussion on memory consumption and compute time to the revised version. This may be an expanded version of responses given to reviewer no2m on these points.


**Note From Pc:**

if the above contains the word "oral" or "spotlight" please see: "oral" presentation means -> notable-top-5% and "spotlight" means -> notable-top-25%. As stated in our emails, we are disassociating presentation type from AC recommendations

**Summary Of Ac-Reviewer Meeting:**

N/A